# Defect-enriched iron fluoride-oxide nanoporous thin films bifunctional catalyst for water splitting

Xiujun Fan [1,2], Yuanyue Liu [3], Shuai Chen [4], Jianjian Shi [3], Juanjuan Wang [5], Ailing Fan [6], Wenyan Zan [7], Sidian Li [7], William A. Goddard III[8] & Xian-Ming Zhang [1,2]

Developing cost-effective electrocatalysts operated in the same electrolyte for water splitting, including oxygen and hydrogen evolution reactions, is important for clean energy technology and devices. Defects in electrocatalysts strongly influence their chemical properties and electronic structures, and can dramatically improve electrocatalytic performance. However, the development of defect-activated electrocatalyst with an efficient and stable water electrolysis activity in alkaline medium remains a challenge, and the understanding of catalytic origin is still limited. Here, we highlight defect-enriched bifunctional eletrocatalyst, namely, three-dimensional iron fluoride-oxide nanoporous films, fabricated by anodization/ fluorination process. The heterogeneous films with high electrical conductivity possess embedded disorder phases in crystalline lattices, and contain numerous scattered defects, including interphase boundaries, stacking faults, oxygen vacancies, and dislocations on the surfaces/interface. The heterocatalysts efficiently catalyze water splitting in basic electrolyte with remarkable stability. Experimental studies and first-principle calculations suggest that the surface/edge defects contribute significantly to their high performance.

[1] Institute of Crystalline Materials, Shanxi University, 030006 Taiyuan, China. [2] Key Laboratory of Magnetic Molecules and Magnetic Information Material of Ministry of Education, School of Chemistry and Material Science, Shanxi Normal University, 041004 Linfen, China. [3] Deparment of Mechanical Engineering and Texas Materials Institute, The University of Texas at Austin, Austin, TX 78712-0292, USA. [4] State Key Laboratory of Coal Conversion, Institute of Coal Chemistry, Chinese Academy of Science, 030001 Taiyuan, China. [5] Scientific Instrument Center, Shanxi University, 030006 Taiyuan, China. [6] College of Materials Science and Engineering, Beijing University of Technology, 100124 Beijing, China. [7] Institute of Molecular Science, Shanxi University, 030006 Taiyuan, China. [8] Materials and Process Simulation Center, The Resnick Sustainability Institute, California Institute of Technology, Pasadena, CA 91125, USA. Correspondence and requests for materials should be addressed to X.F. (email: fxiujun@gmail.com) or to X.-M.Z. (email: xmzhang@sxu.edu.cn)

Splitting water via electricity provides an attractive method to efficiently produce renewable energy. This process can be divided into the anodic oxygen evolution reaction (OER) and the cathodic hydrogen evolution reaction (HER) processes[1]. Notably, in energy production device applications, water electrolyzers typically employ catalysts, consisting of precious metals and their oxides (Pt, $RuO_2$, $IrO_2$) as catalysts to faciliate these half-reactions[2]. The scarcity nature of these catalysts considerably impede their large-scale commercial utilization. Therefore, developing new electrocatalysts with both good activity and low cost is highly imperative. Recently, substantial efforts have been made to explore highly active non-noble metal catalysts such as transition metal sulfides[3]/phosphides[4]/nitrides[5]/carbides[6]/selenides[7]/borides[8] for HER in acidic electrolytes, and transition metal oxides[9]/hydroxides[10]/nitrides[11] for OER in basic electrolytes. Nevertheless, owing to the sluggish four-proton-coupled electron transfer and rigid oxygen–oxygen bonding, the integration of overall water splitting is still constrained by the bottleneck OER process. Furthermore, for practical applications, a water splitting electrolyzer with dual ability require operation in the same electrolyte, especially in strongly alkaline medium. Hence, developing inexpensive HER and OER bifunctional catalysts with efficient activity toward water electrolysis becomes important yet challenging.

Iron compounds, particularly oxide ($Fe_2O_3$) and fluoride ($FeF_2$) have received attention for decades owing to their abundance, stability, and environmental compatibility, resulting high potential in supercapacitors[12], sodium-ion batteries[13], lithium batteries[14,15], photocatalysts[16], and electrocatalysts[17]. Unfortunately, the electroconductibility of iron oxide and fluoride is poor, which greatly limits their wide-spread application; iron oxide is an indirect band gap semiconductor[18], while iron fluoride has low conductivity due to the higher ionicity of Fe–F bonds[19]. Unlike iron oxide and fluoride, iron fluoride-oxide phases have high electroconductivity as the coexistence of Fe–O and Fe–F bonds[14,20]. Meanwhile, the overall conductivity in nanomaterials especially nanoparticles is significantly limited on account of the undesirable intergranular electron transport, which slows the total electrocatalytic activity[21]. Additionally, both experimental and computational studies concluded that defects play a crucial role in electrocatalytic catalysis[22], and thus increase of defect content favors the electrocatalytic activity[23]. Three-dimensional (3D) ultrathin nanoporous films have remarkable electroactive surface/active sites, and can decrease the ionic diffusion length and increase the contact area with electrolyte[24]. Hence, 3D iron fluoride-oxide nanoporous film (IFONF) with both abundant defect sites and high electroconductivity would be a promising bifunctional water splitting catalyst. However, owing to the demanding conditions required for synthesis (highly toxic materials[20], expensive precursors[25], and complicated processes[26]), IFONF-based electrocatalyst with high activity and durability is still rare.

Herein, a facile low temperature synthesis is developed to fabricate defect-enriched IFONFs, namely, partial conversion of the nanoporous Fe-oxide into iron fluoride-oxide through reaction with fluorine vapor (from ammonium fluoride, $NH_4F$). In IFONFs, modulations of defect states including interphase boundary, phase junction architecture, and so on, are accomplished by controlling the extent of iron oxide to fluoride-oxide phase transformation. The resulting IFONFs with high electrical conductivity possess abundant defect states on surfaces/interface, which expose additional reaction sites and lower the adsorption energy of the reactant and product. IFONFs demonstrate a superb HER and OER activity in basic electrolyte with impressive stability. This work represents the first synthesis of $FeF_2$–$F_2O_3$ heterocatalysts via an easy low temperature anodization/

fluorination strategy, which opens up a low cost and scalable route to fabricate transition metal fluorides-based materials for application in advanced fields.

## Results

**Synthesis and characterizations of IFONFs.** A typical fabrication procedure is illustrated in Fig. 1a along with scanning electron microscope (SEM) images (Fig. 1b–e). Firstly, anodic treatment of Fe foil was performed potentiostatically to achieve Fe-oxide porous thin film (PTF) with pore sizes around 40 nm in average (Fig. 1b, Supplementary Fig. 1). Then, in a chemical vapor deposition (CVD) apparatus, the anodized thin film was reacted with hydrogen fluoride vapor (from $NH_4F$) at 300–400 ° C. Further details of the experiments are provided Supplementary Notes 1 and 2. The fabrication process is time-saving and non-toxic, with iron-based nanoporous film directly grown on Fe thin foil (Supplementary Fig. 2) that has excellent flexibility and low cost. The resulting films, namely, IFONFs-$x$, where $x$ denotes the fluorinated time ($T_{\text{fluorinated}}$) in unit minute. With $T_{\text{fluorinated}}$ of 15 min, the CVD fluorinated PTF exhibits nanopores structure (Supplementary Fig. 3a). The layers are gradually crystallized to more ordered (IFONFs-30, Supplementary Fig. 3b) and converted into a uniform nanoporous film with open pores (IFONFs-45, Fig. 1c). IFONFs-45 inherits the nanoporous nature and 3D morphology from the as-anodized Fe-oxide PTF without damaging the ordered porosity, revealing an open and porous framework (Fig. 1d). Further extending $T_{\text{fluorinated}}$ to 90 min does not cause continuous growth of iron fluoride-oxide nanocrystals (NCs), and the obtained films still maintain the self-organized high-porosity nature with grainy porous wall (Supplementary Fig. 3c). The cross-sectional SEM image (Fig. 1e and Supplementary Fig. 4) displays a nanochannel structure throughout the layer, where the internal surface of nanoporous film was exposed to the electrolyte. The IFONFs with thick of ~2.5 μm (Fig. 1e, mass loading ~0.2 mg cm$^{-2}$) were then directly utilized in water electrolysis.

X-ray diffraction (XRD, Fig. 1f) and X-ray photoelectron spectroscopy (XPS, Fig. 1g–i, Supplementary Figs. 5 and 6 and Supplementary Discussion 1) were used to identify the chemical conversions from Fe-oxide to iron fluoride-oxide. The surface atomic concentrations obtained with XPS (Supplementary Fig. 5) and the elemental compositions determined by inductively coupled plasma mass spectrometry (ICP-MS) are concluded in Supplementary Table 1. For Fe 2$p$ XPS (Fig. 1g), iron atoms possess three mixed valence states including $Fe^{3+}$, $Fe^{2+}$, and $F^0$ in both Fe-oxide and IFONFs-45. After deconvolution, the fitted peaks around 710.7 and 723.9 eV are designated as $Fe^{3+}$ 2$p_{3/2}$ and $Fe^{3+}$ 2$p_{1/2}$ of $Fe_2O_3$, respectively; those around lower binding energies (713.2 and 716.5.3 eV) and higher binding energies (726.6 and 729.9 eV) correspond to $Fe^{2+}$ 2$p_{3/2}$ and $Fe^{2+}$ 2$p_{1/2}$ states of $FeF_2$, respectively[13], clearly indicating the heterostructure is a mixture of $FeF_2$ and $Fe_2O_3$. A similar result is also seen in O 1$s$ XPS spectra of IFONFs-45. As revealed in Fig. 1h, the broad fitted peak located at around 529.9 eV corresponds to Fe–O, which is a typical peak for $Fe_2O_3$ that has been exposed to fluoride[27], while weak peak at ~532.1 eV is assigned to C = O groups[28]. The F 1$s$ peak with binding energy of 684.4 eV is attributed to Fe–F, reflecting a similar fluorination state of F in iron fluoride-oxide to that of $FeF_2$[13] (Fig. 1i). Therefore, the above results confirm the successful transformation of Fe-oxide into iron fluoride-oxide. Additionally, $Fe_2O_3$ PTF was prepared according to our previous study[29] for comparison (XPS and XRD data are given in Supplementary Fig. 7), and also successfully converted into $FeF_2$ PTF with fluorination (Supplementary Fig. 8). IFONFs are also prepared at a fixed $T_{\text{fluorinated}}$ (45 min) with

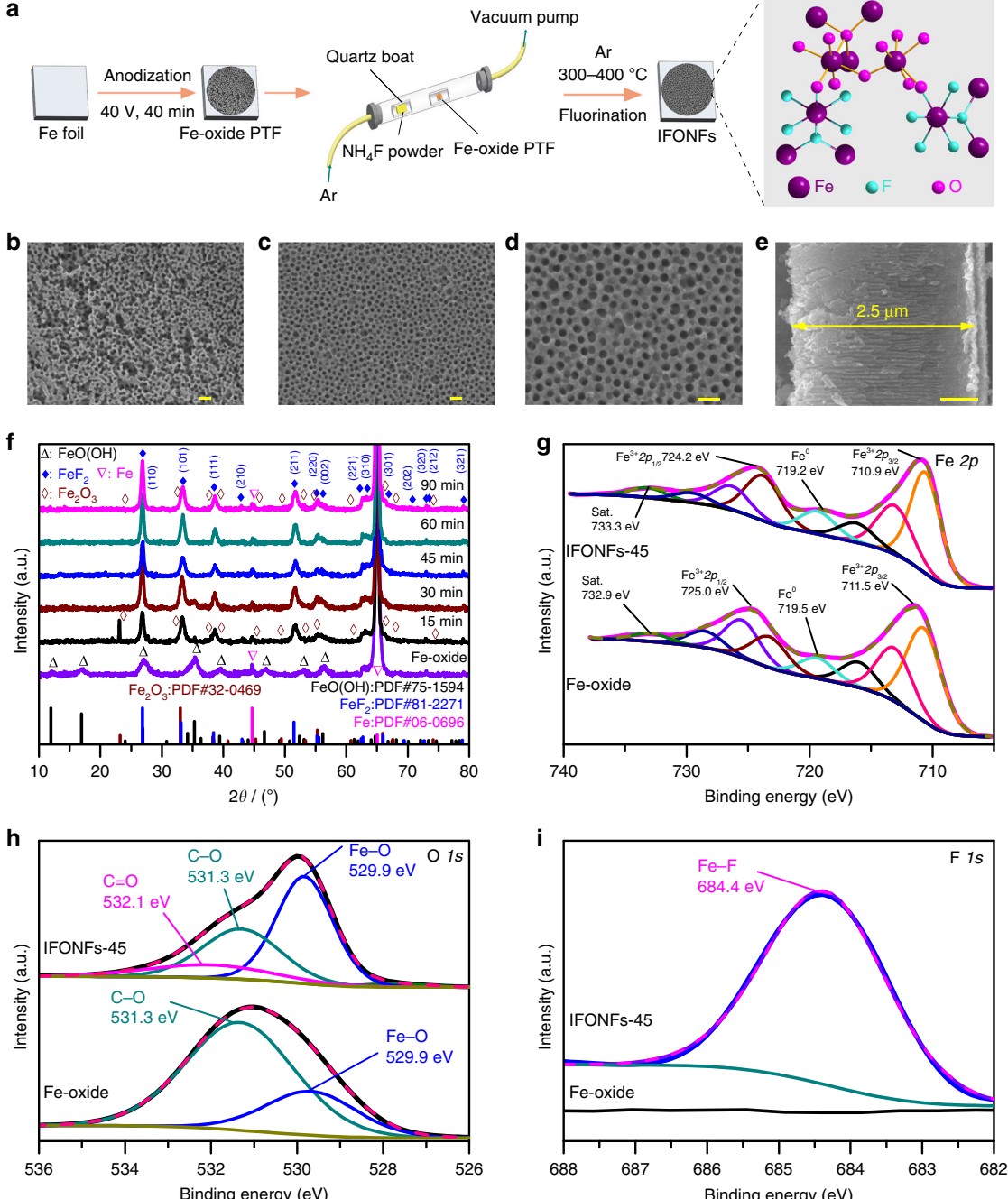

**Fig. 1** IFONFs fabricated from commercial Fe foil with anodization/fluorination process. **a** Schematic of the fabrication process for IFONFs, starting with a Fe foil. SEM top-view images of **b** Fe-oxide PTF and **c** IFONFs-45. Scale bar, 100 nm. **d** High-magnification SEM image of IFONFs-45. Scale bar, 100 nm. **e** SEM cross-sectional image of IFONFs-45. Scale bar, 500 nm. **f** XRD spectrum of Fe-oxide PTF and IFONFs synthesized with various $T_{fluorinated}$. For Fe-oxide PTF, the diffraction peaks at $2\theta = 11.9°$, 16.9°, 26.9°, 35.3°, 39.4°, 46.6°, 53.1°, and 56.2° suggests the formation of FeO(OH) (PDF#75–1594) with anodization. The Fe-oxide PTF anodically grown on Fe substrates consists of FeO(OH) and Fe, while the Fe peaks are from Fe foil. With short $T_{fluorinated}$ of 15 min, the diffraction peaks of both $Fe_2O_3$ and $FeF_2$ are sharp and intense, and $Fe_2O_3$ is obtained by annealing FeO(OH) in an argon atmosphere during fluorination[47]. With $T_{fluorinated}$ of 45 min, the primary diffraction peaks associated with $Fe_2O_3$ are declined while both peaks for $FeF_2$ (110) and (101) are distinguished, indicating a $T_{fluorinated}$-dependent phase transformation for iron fluoride-oxide in fluorination condition. $Fe_2O_3$ and $FeF_2$ could coexist with fluorination, and $FeF_2$ phase has a tetragonal structure with a P42/mnm space group (PDF#81–2271). High-resolution XPS in **g** Fe $2p$, **h** O $1s$, and **i** F $2p$ regions for Fe-oxide PTF and IFONFs-45, respectively. For IFONFs-45 in **g**, two core-level signals of Fe $2p$ located at ~710.9 and 724.2 eV are attributed to Fe $2p_{3/2}$ and Fe $2p_{1/2}$, which are negatively shift ~0.6 and ~0.8 eV relative to those in the raw Fe oxide, respectively, owing to the bond formation of Fe and F elements. The satellite peaks at ~719.9 and 733.3 eV are attributed to $Fe^0$, suggesting the presence of elemental Fe on the surface. Moreover, the O $1s$ peak of IFONFs-45 in **h** is shifted to lower binding energy with respect to the Fe-oxide PTF, which attributes to the formation of $FeF_2$ phase

different temperature. As revealed in Supplementary Figs. 9 and 10, the fluorination temperature has an influence on the morphology and pore size of nanoporous film. XPS illustrates clear signals for F 1s, Fe 2p, and O 1s in IFONFs fluorinated at 300–400 °C (Supplementary Fig. 11), indicating the temperature has no significant effect on the crystalline type. These characterizations confirm that a fluoride-oxide nanostructure is obtained via fluorination reaction in the quartz socket tube with $NH_4F$.

Further analysis on the morphology of IFONFs was then carried out adopting transmission electron microscopy (TEM). As presented in Fig. 2a and Supplementary Fig. 12, interconnected network of IFONFs-45 with ordered nanopores are maintained with fluorination process. High-resolution TEM (HRTEM) analyses reveal that all nanopores consist of openings around 40 nm with thin amorphous region on the inner wall (Fig. 2b and Supplementary Fig. 13a). Figure 2c represents the zoom-in imaging of a rectangular region in Fig. 2b (marked with yellow-dotted line), which indicates that the nanopores possess a smooth space morphology with amorphous rims containing a small amount initiation remnant or other etching artifacts[30]. The amorphous nanodomain (marked by white-dashed line, Fig. 2c, and Supplementary Fig. 13b) is the residuum of as-formed Fe-oxide nanoporous films. From the cross-section TEM imaging of Fig. 2d, the aligned nanopores have tubular structures with 40 nm in diameter, and are oriented perpendicular to substrate surface, coinciding with SEM observation (Fig. 1e). The element contents and distribution in the matrixes were identified based on both energy-dispersive X-ray spectroscopy (EDS, Supplementary Fig. 14) and elemental mapping (Supplementary Fig. 15). The average atomic ratio of F/O is ∼0.3 calculated from the EDS spectrum, giving a ∼30% substitution of O sites. Scanning TEM (STEM) image reveals that IFONFs-45 possesses a highly porous texture and interconnected structure throughout the whole heterostructure, while EDS mapping images verify a homogeneous distribution of Fe, F, O, and C elements (Fig. 2e).

Structure evolutions and defect states occur when cubic iron oxide meets tetragonal iron fluoride, and $T_{fluorinated}$ evidently impacts the extent of iron oxide to fluoride phase transformation, which in turn determines the defect states. Atomic-resolved TEM and fast Fourier transform (FFT) images were used to visualize defect states in IFONFs. With short $T_{fluorinated}$ (i.e., 15 min), $FeF_2$ (separated by white-dotted lines) and $Fe_2O_3$ nanodomains (separated by yellow-dotted lines) are located in randomly orientation, demonstrating the deficiency of defects (Fig. 3a and Supplementary Fig. 16). With fluorination processing for 30 min, the $FeF_2$ and $Fe_2O_3$ nanodomains have several sub-nanostructures, are separated distributed on supports with less defect (Fig. 3b and Supplementary Fig. 17). Further extending $T_{fluorinated}$ to 45 min, the morphology and defect states of IFONFs begin to change. The IFONFs-45 heterogeneous matrixes exhibit a relatively smooth surface and contain well-bonded phase junctions (Fig. 3c), indicating strong coupling interactions and interface reconstruction inbetween $Fe_2O_3$ and the grafted $FeF_2$ nanodomains. The edges of nanodomains and a perfect match between $FeF_2$ and $Fe_2O_3$ phases are easily visible (Fig. 3d). The neighboring nanodomains merge together at the boundaries free of visible gaps, indicating the strong connections between $FeF_2$ and $Fe_2O_3$, ensuring stable electrical and mechanical contact (Fig. 3e and inset). To further inspect the planar defects generated on the heterogeneous matrixes, aberration-corrected STEM images were conducted (Fig. 3f, g and Supplementary Fig. 18). Besides dense nanograin boundaries, several lattice scale alterations are also induced in the heterostructure. As illustrated in Fig. 3g, the $Fe_2O_3$ nanodomain exhibits an extended and disordered defect structure with interlayer spacing of 0.248 nm. This is slightly larger than the $d$ spacing of (400) planes of cubic $Fe_2O_3$ (marked as i). Whereas, $FeF_2$ displays continuous lattice fringes with spacing of 0.325 nm ((110) atomic plane, marked as j); this spacing is smaller than the interlayer spacing of 0.332 nm in bulk $FeF_2$. Such significant lattice distortions on basal surface confirm the presence of numerous dangling bonds around the Fe centers, which increase the intrinsic catalytic activity of active sites[31]. In the FFT images, the obtained splitting in diffraction spots marked by arrows further verifies the existence of imperfections in IFONFs-45 along some crystallographic orientations (Fig. 3i, j). The atomic-scale features in $FeF_2$ nanodomains are the planar defects in the form of stacking faults (SFs), which nucleated on different (101) planes to form SF cross-structures (marked as h in Fig. 3g, and Fig. 3h), revealing (101) planes have the lowest energies for SFs formation during CVD fluorination[32]. The $FeF_2$ nanodomains display some tetragonal lattice area (Fig. 3h), while the $Fe_2O_3$ nanodomains reveal trigonal lattice area (octahedral coordination) with the common honeycomb lattice of

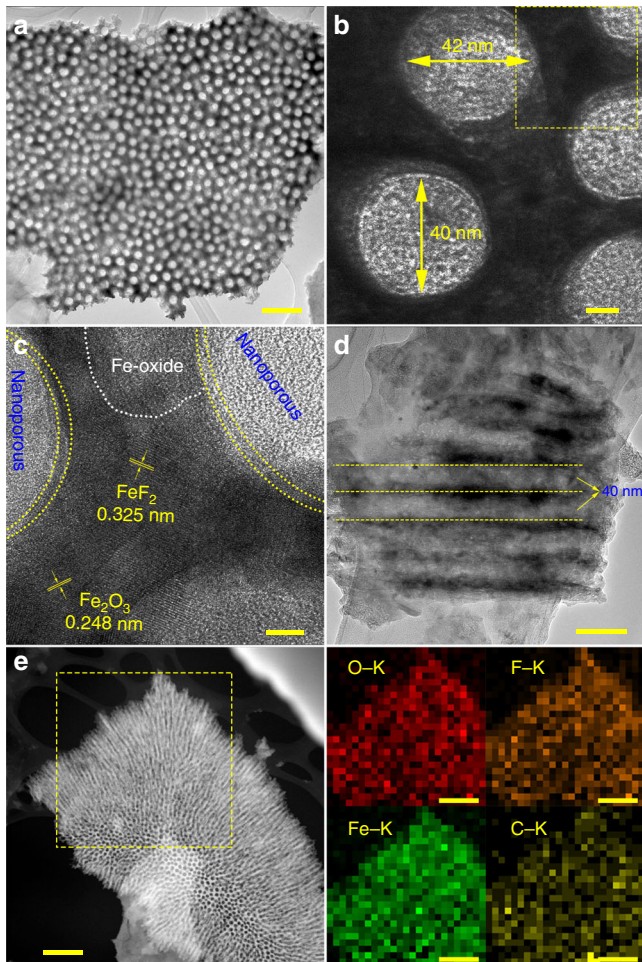

**Fig. 2** Microscopic morphology and chemical composition of IFONFs-45. **a** TEM top-view images of IFONFs-45. Scale bar, 200 nm. **b, c** HRTEM images of IFONFs-45. Besides amorphous nanodomains, the lattice fringes with spacing of ∼0.325 and 0.248 nm ascribed to $FeF_2$ (110) and $Fe_2O_3$ (400), respectively, are clearly visible, indicating crystallinity of iron fluoride-oxide nanostructure. The nanopores, together with fine nanograins of iron fluoride-oxide and thin amorphously shells are uniformly distributed in the porous network, ensuring IFONFs-45 has more accessible sites for electrochemical reactions. Scale bar, 10 and 5 nm, respectively. **d** TEM cross-sectional image of IFONFs-45. Scale bar, 100 nm. **e** STEM elemental mapping of IFONFs-45. Scale bar, 500 nm

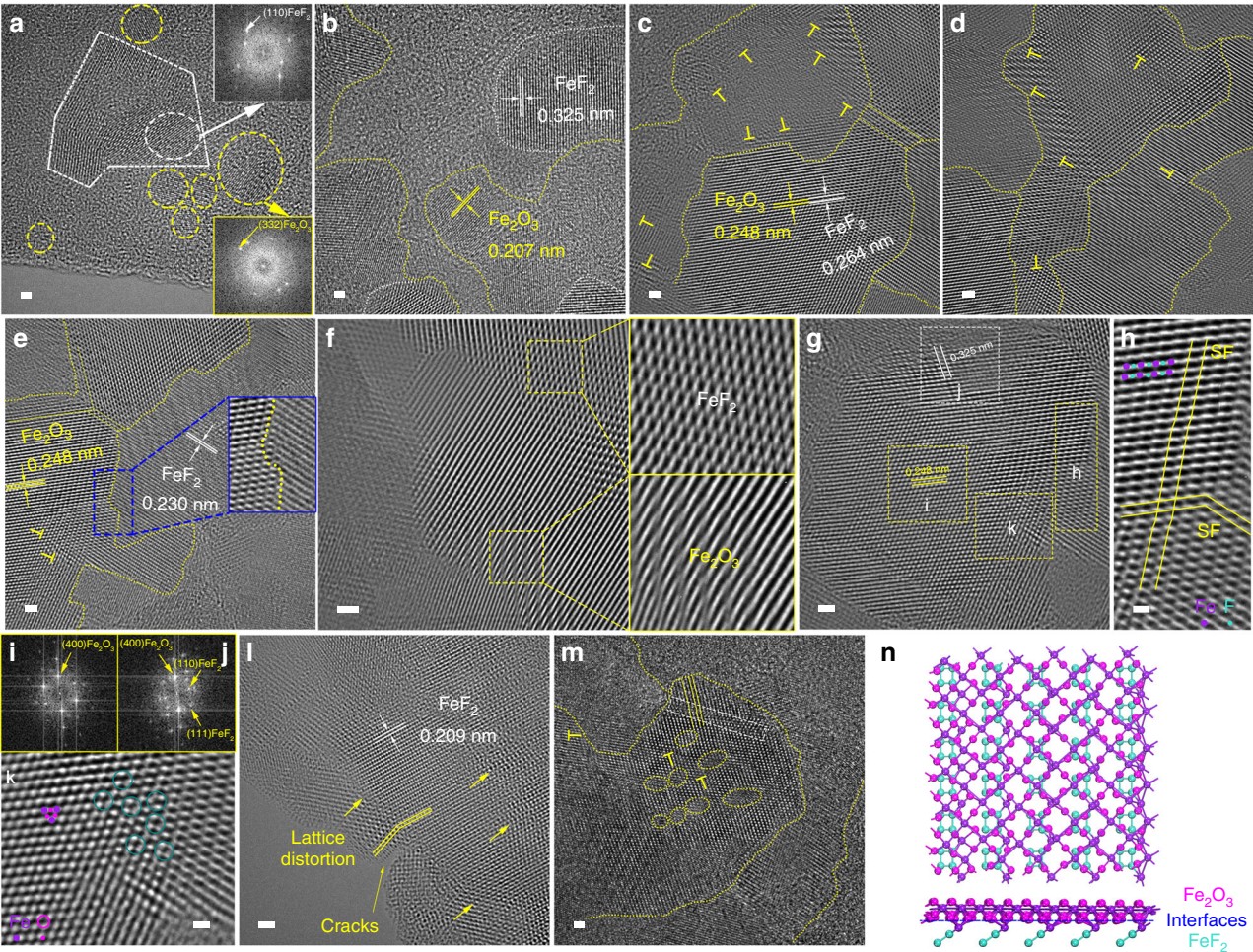

**Fig. 3** Atomic-resolution structure analyses of IFONFs samples. **a–e** HRTEM images and corresponding FFT patterns of IFONFs with various $T_{fluorinated}$ for **a** 15 min, **b** 30 min, and **c–e** 45 min. The IFONFs-45 heterostructures in **c–e** consist of numerous dislocations and distortions marked with "T", and maintain the original two phases arrangement with nanodomain structure. **f** STEM images for IFONFs-45, revealing continuous lattice fringes (top right for (101) of FeF$_2$ and bottom right for (400) of Fe$_2$O$_3$). Moiré patterns are observed in FeF$_2$ and Fe$_2$O$_3$ nanograins, with slight rotation in the phase junction architecture[48], which suggests a defect-rich heterogeneous structure. **g** STEM image and **i, j** the corresponding FFT patterns of iron fluoride-oxide nanodomains at selected regions. **h, k** The zoom-in image of yellow-boxed region in **g** depicts the detailed structure of stacking faults and vacancy-type defects, respectively. The O vacancies in Fe$_2$O$_3$ lattice in **k** are highlighted by darkcyan circles. **l, m** Crack and distortions in iron fluoride-oxide nanodomains. Apart from microstructural imperfections (yellow rectangle) and dislocations (yellow circles) in **m**, IFONFs-45 also has lattice imperfections (yellow "T") along with few overlapped planes, illustrating the defective heterostructure consists of amorphous and crystalline planes. **n** Atomic structural models of FeF$_2$–F$_2$O$_3$ hybrid, top and bottom parts are from side and top view, respectively. FeF$_2$ (101) surface has fluorine or iron termination, and Fe$_2$O$_3$ (400) surface has oxygen or iron termination; Fe$_2$O$_3$ (400) surface with oxygen termination is more reactive than iron termination to iron termination of FeF$_2$ (101) surface, and thus the FeF$_2$–F$_2$O$_3$ heterostructure can be constructed. Accordingly, the O atoms in the Fe$_2$O$_3$ matrix could bond with Fe atoms in the FeF$_2$ matrix, leaving the about four O atoms unsaturated per Fe$_2$O$_3$ unit cell, namely, the bonding number of O atoms in bulk Fe$_2$O$_3$ subtracts the bonding number of O atoms in Fe$_2$O$_3$ (400) and FeF$_2$ (101) surfaces, because of the relatively less ratio of Fe atoms to other atoms in FeF$_2$ than Fe$_2$O$_3$. Turquoise = F, purple = Fe, magenta = O. Scale bar, 1 nm in **a–g** and **l–m**, 0.3 nm in **h, k**

the trigonal prismatic coordination in the cubic phase (Fig. 3k). Interestingly, IFONFs-45 displays oxygen vacancies ($V_O$)-type point defects (Fe$_2$O$_3$ nanodomain, marked as **k** in Fig. 3g, and Fig. 3k). The presence of $V_O$ can decrease the adsorption energy of intermediates (e.g., OH*, O*, and OOH*) at active sites and facilitate the adsorption of these intermediates[33]. To validate the above assumption, we have performed additional calculations for OH adsorption on perfect and defective Fe$_2$O$_3$ (with oxygen vacancy), as shown in the Supplementary Fig. 19. In addition, F insertion causes short cracks together with lattice distortions at the edge of continuous basal plane (labeled by yellow arrows, Fig. 3l). The cracks on surface dramatically boost surface area, which would offer more coordinatively unsaturated sites for catalytic reactions (Supplementary Fig. 20); while the lattice

rotations and discontinued crystal fringes are also locally observed in the nanoporous films (Supplementary Fig. 21), highlighting the existence of numerous defects. Despite multiple defects, the iron fluoride-oxide hybrids similarly possess a tetragonal FeF$_2$ phase. Figure 3m represents FeF$_2$–F$_2$O$_3$ nanodomains separated by an amorphous grain boundary and defect region (marked by a set of yellow-dotted lines). The enhanced surface/edge defect state is believed to increase the exposure of active basal-plane/edge sites, which is beneficial to electrocatalysis.

The defect states of IFONFs can be tailored by $T_{fluorinated}$. HRTEM images of IFONFs samples with $T_{fluorinated}$ for 60 and 90 min are presented in Supplementary Figs. 22 and 23, suggesting the declined defects. Further fluorination does not significantly

increase the sizes of the interconnected $FeF_2$–$F_2O_3$ nanodomains, which is in accordance with SEM images (Supplementary Fig. 3c). But, more F ions incorporated into iron oxide nanodomains minimize the concentration of iron fluoride-oxide heterostructure, giving a reduced defect amount. Therefore, the extraneous F ions incorporating into $Fe_2O_3$ nanodomains gives rise to atomic defects, which can be modulated by controlling the extent of iron oxide to fluoride phase transformation. The $T_{fluorinated}$ is essential for the phase transformation of IFONFs, which is in accordance with the XPS results (Supplementary Fig. 5). Taken together, the local structural information on $FeF_2$–$F_2O_3$ heterostructure can be illustrated as Fig. 3n, Supplementary Fig. 24, and Supplementary Discussion 1. As displayed in local structure, a $3 \times 3$ (400) surface of $F_2O_3$ is used to match a $5 \times 6$ (101) surface of $FeF_2$, and the lattice mismatch of the $F_2O_3$ and $FeF_2$ relative to $F_2O_3$ is around 1.9%, indicating the structure is stable, which is also in accordance with HRTEM results in Fig. 3c, f where (400) planes of $Fe_2O_3$ are parallel to (101) planes of $FeF_2$.

**Electrocatalytic HER and OER performances**. To assess the HER electrocatalytic activity, the as-prepared IFONFs catalyst were directly used as working electrodes and evaluated in $H_2$-saturated 1 M KOH electrolyte (see Supplementary Notes 3 and 4 for experimental details, Supplementary Fig. 25). Commercial Pt and the raw Fe-oxide were also measured for comparison. The appealing HER activities of IFONFs are evaluated by the linear sweep voltammetry (LSV) curves and corresponding Tafel plots. As illustrated in Fig. 4a, the IFONFs catalysts possess much lower onset potentials ($\eta$) than that of raw Fe-oxide PTF; among all IFONFs-45 affords the lowest value of 20 mV, indicating that the IFONFs-45 decreases the reaction kinetic barrier for HER. IFONFs-45 yields an extremely large cathodic current density of 101.4 mA cm$^{-2}$ at $\eta = 200$ mV, which is 181 times larger than that of Fe-oxide PTF and much larger than that of other four fluorinated materials (2.3–84.9 mA cm$^{-2}$). The exchange current density ($j_0$) of 0.0950 mA cm$^{-2}$ for IFONFs-45 outperforms the values of 0.0470 mA cm$^{-2}$ for IFONFs-30 and 0.0322 mA cm$^{-2}$ for IFONFs-60, again suggesting the excellent HER inherent activity of IFONFs-45. The corresponding Tafel plots indicate that IFONFs-45 possesses smaller Tafel slopes (31 mV dec$^{-1}$) than that of less defect catalyst (68 mV dec$^{-1}$ for IFONFs-30, and 55 mV dec$^{-1}$ for IFONFs-60), and verifies efficient kinetics of $H_2$ evolution (Fig. 4b). In short, the optimal $T_{fluorinated}$ in terms of the HER activity is determined to be 45 min. Bare $Fe_2O_3$ and $FeF_2$ PTF present poor HER activity in terms of the largest Tafel slopes (154 and 235 mV dec$^{-1}$, respectively) (Supplementary Fig. 26 and Supplementary Table 2), highlighting the cooperative interactions between $Fe_2O_3$ and $FeF_2$ in IFONFs-45 electrocatalyst. These experiments conclude that the HER activity of IFONFs can be modulated by their defect states. The HER activities of IFONFs prepared with various fluorinated temperature in control experiments were also measured (Supplementary Fig. 27). Compared with Fe-oxide, $Fe_2O_3$, $FeF_2$, and Fe-based electrocatalysts reported recently, the defective IFONFs-45 delivers lower Tafel slopes and larger kinetic current densities (Supplementary Tables 2 and 3). This suggests that similar to the $MoS_2$[22], introduction of defects to create active sites in basal plane is an effective way to improve the catalyst activity. Based on the electrochemical double-layer capacitance ($C_{dl}$) measurements via cyclic voltammetry at different scan rates (Fig. 4c and inset), the electrochemically active surface area can be determined as well. The $C_{dl}$ of IFONFs-45 is 63.40 mF cm$^{-2}$, which is much larger than that of IFONFs with other $T_{fluorinated}$ (Supplementary Fig. 28), whereas bare $Fe_2O_3$ and $FeF_2$ PTF have low $C_{dl}$ of 16.28 and 13.38 mF cm$^{-2}$, respectively (Supplementary

Fig. 29). As a result, the IFONFs-45 yields an active surface area of 18.5, 3.9, and 4.7 times larger than that of raw Fe-oxide, $Fe_2O_3$, and $FeF_2$ electrodes, respectively, suggesting the fluorination process can improve the active electrocatalytic area and make more active sites exposed. Indeed, the roughness factor ($R_F$) dramatically increases from 328.3 (IFONFs-15) to 1585.0 (IFONFs-45) with fluorination process (for calculation details, see Supplementary Note 4). The increased surface area and atomic defects are considered to contribute to the improvement in HER performance. Based on an electrochemical method, the number of active sites was quantified (Supplementary Fig. 30). IFONFs−45 gives the number of active sites of $1.09 \times 10^{-6}$ mol, much larger than that of IFONFs−30 ($1.94 \times 10^{-7}$ mol) and IFONFs−60 ($2.26 \times 10^{-7}$ mol). The activity of IFONFs was further evaluated in terms of the apparent turnover frequencies (TOFs), assuming that all Fe ions in the nanoporous films act as active sites (Supplementary Note 5). IFONFs-45 affords the highest TOF value of 0.27700 $H_2$ s$^{-1}$ at $\eta = 100$ mV, revealing that IFONFs-45 delivers higher activity than other iron fluoride-oxide catalysts[34]. In addition, to further insight into HER kinetics, the electrochemical impedance spectroscopic analysis was employed at $\eta$ of 5 mV (Fig. 4d and inset). A low charge-transfer resistance ($R_{ct}$) of 7.4 Ω is observed for IFONFs-45, suggesting a facile HER kinetics at the electrode–electrolyte interface (Supplementary Discussion 1).

Given the high HER activity of defect-riched IFONFs, we further analyzed their stability and durability to continuously catalyze the generation of $H_2$. As presented in Fig. 4e, IFONFs-45 retains stable current density over 30,000 s of continuous operation at different potentials with negligible changes, even though a high potential of −172 mV is adopted. In contrast, the current density of Pt/C decreases from 82 to 74.1 mA cm$^{-2}$ for 30,000 s of continuous operation (Supplementary Fig. 31). This confirms the higher stability of IFONFs-45 than that of Pt/C. The LSV curves of IFONFs-45 in 1 M KOH before and after 1000 and 3000 cycles deliver small cathodic shifts of ~9 and ~10 mV to reach −10 mA cm$^{-2}$, respectively, proving a great durability (Fig. 4f). Excellent structural integrity of IFONFs-45 is confirmed by HRTEM images taken after the durability test (Supplementary Fig. 32), which further confirms the robustness of the hybrid catalyst.

OER kinetics of IFONFs was also carefully investigated. For comparison, the OER performance of commercial $RuO_2$ deposited on carbon paper was also tested. As displayed in Fig. 4g, with $T_{fluorinated}$ increase from 15 to 45 min, onset potential gradually decrease from 1.44 to 1.39 V vs RHE, implying an initial enhanced OER activity with increase of iron fluoride phase. However, when $T_{fluorinated}$ is extended to 90 min, the onset potential increases to 1.42 V vs RHE instead, indicating the further increase of iron fluoride phase downgrades the OER activity. In addition, IFONFs-45 gives higher specific current density than that of Fe-oxide PTF and IFONFs with other $T_{fluorinated}$, and emerges a small onset potential, which confirms that defect-riched IFONFs-45 owns a superior intrinsic catalytic activity toward OER. IFONFs-45 affords a much lower $\eta_{10}$ (the $\eta$ required at the current density of 10 mA cm$^{-2}$, 1.49 V vs RHE) than that of the counterparts with other $T_{fluorinated}$, such as 15 min with 1.61 V, 30 min with 1.52 V, and 60 min with 1.51 V (Fig. 4h, left). Different Tafel slopes are identified from IFONFs-45 (45 mV dec$^{-1}$), Fe-oxide PTF (207 mV dec$^{-1}$), and $RuO_2$ (125 mV dec$^{-1}$), which imply different rate-determining steps for a given pathway (Fig. 4h right). The decrease in Tafel slope from 86 to 45 mV dec$^{-1}$ for the IFONFs fluorinated from 15 to 45 min may be caused by a change in the rate-determining step from a three electron transfer to a two electron transfer[35]. In comparison to the less defect counterparts, IFONFs-45 exhibits smaller Tafel

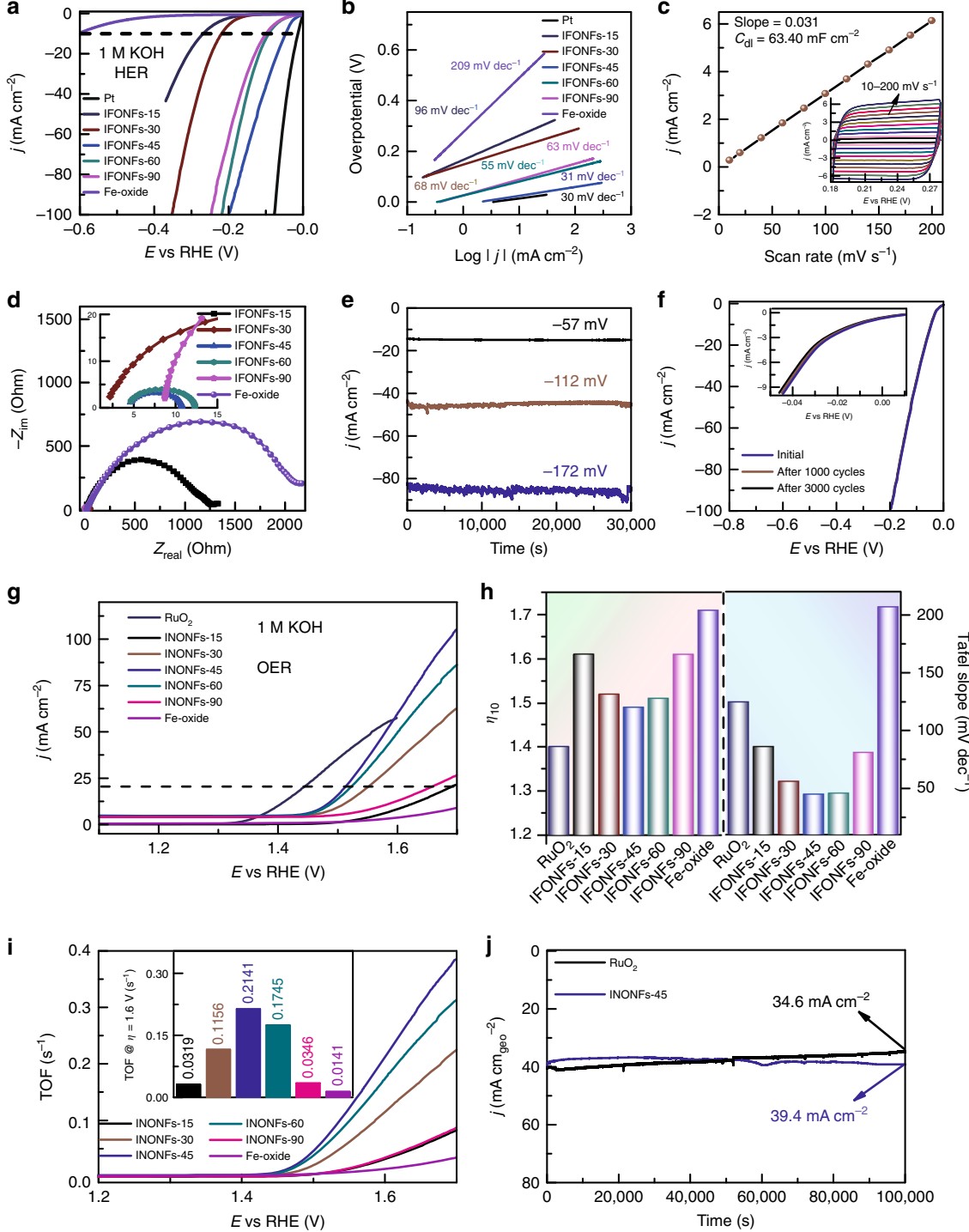

**Fig. 4** Electrochemical HER and OER activity of IFONFs. **a** Polarization curves measured at 50 mV s$^{-1}$ in 1 M KOH aqueous solution, and **b** the corresponding Tafel slopes. **c** The capacitive current at 0.23 V (vs RHE) as a function of scan rate for IFONFs-45 ($\Delta j_0 = j_a - j_c$). The inset is CVs for IFONFs-45 with different rates from 10 to 200 mV s$^{-1}$. **d** Nyquist plots measured at a voltage −5 mV (vs RHE) over the frequency range 1000–0.01 Hz in 1 M KOH. The inset is enlarged area denoted by dash squares. **e** CV cycle-dependent current density at different potentials with 30,000 s in 1 M KOH aqueous solution. **f** Polarization curves before and after 3000 CV cycles of IFONFs-45 ranging from 0 to 0.2 V vs RHE. **g** LSV curves measured at 5 mV s$^{-1}$ in 1 M KOH aqueous solution, **h** the corresponding $\eta_{10}$ (left) and Tafel slopes (right) for RuO$_2$, IFONFs and Fe-oxide catalysts. **i** TOF per oxidative iron site for IFONFs and Fe-oxide catalysts. The inset reveals the TOF values at $\eta = 1.6$ V. **j** Time-dependent current density curve of IFONFs-45 and RuO$_2$ at a fixed overpotential of 1.56 and 1.52 V to drive 40 mA cm$^{-2}$, respectively. dec, decade

slope and lower overpotential, which could be credited to the increased active sites and reduced charge-transfer resistance (Supplementary Fig. 33 and Supplementary Table 2), further confirming the feasibility of electrochemically catalyzing OER at defect-rich electrodes. Additionally, the individual Fe$_2$O$_3$ and FeF$_2$ PTF exhibit lower OER activity than IFONFs (Supplementary Fig. 34), implying that synergistic effects of Fe$_2$O$_3$ and FeF$_2$ could be crucial in enhancing OER activity by influencing the

iron fluoride-oxide hybrid catalyst. The comparison of electro-chemical performance implies that the OER activity in IFONFs-45 hybrid primarily originates from the defect states of iron

fluoride-oxide NCs. The defect enrichment of IFONFs is modulated by $T_{fluorinated}$, which in turn significantly alters the OER activities (Supplementary Discussion 2). Control

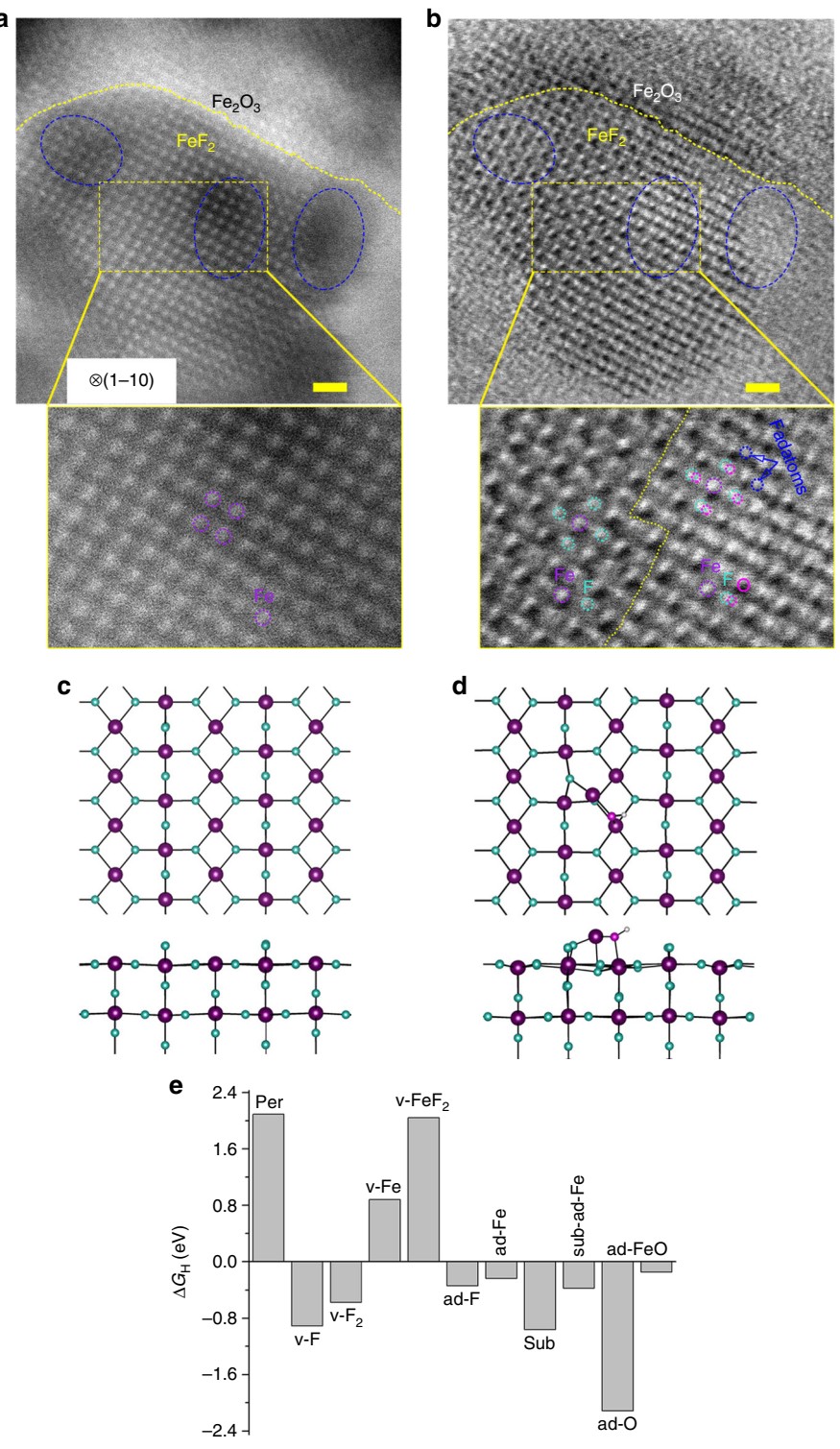

**Fig. 5** Typical atomic-level STEM images and DFT calculations of defect-riched IFONFs-45. **a** HAADF-STEM image of IFONFs-45 taken from [1–10] direction of $FeF_2$. The enlarged views of exact objects in the dashed-box depict detailed distribution of Fe atoms. **b** BF-STEM image of IFONFs-45. The $FeF_2$ tetragonal lattice consisting of Fe (purple dots) and alternating F (turquoise dots) atoms is readily distinguished in left side of dashed-box. The neighboring columns depict faint O signals (magenta dots) at the right of interface, as Fe and F atoms are brighter than the substituted O atoms in bright-field mode. Atomic structure of $FeF_2$ (110) surface, with stoichiometric termination (**c**), and adsorbed with O, Fe, and H (**d**). Fe: purple; F: turquoise; O: magenta; H: white. Both top and side views are revealed. **e** Free energy of H adsorption on different sites. "per" means perfect (stoichiometric) surface, "v" represents vacancies, "ad" for adatoms, "sub" for F substituted by O, and "sub-ad-Fe" for Fe adatom on O that substituted F. Scale bar, 1 nm

experiments (Supplementary Fig. 35) displays that IFONFs fluorinated with 45 min at 300 and 400 °C have low activity with Tafel slopes of 66 and 119 mV dec$^{-1}$, respectively, which is in connection to their morphology (Supplementary Fig. 9). Hence, the reduced Tafel slope indicates better catalytic behavior with superior type of the active sites in IFONFs-45. Concerning the low onset potential, lowered $\eta_{10}$, and small Tafel slope, it is found that the catalytic activity of IFONFs-45 surpasses the state-of-the-art OER catalysts (for detailed comparison, see Supplementary Table 4).

To further unravel the intrinsic activities, the number of active sites and TOF for OER were calculated on basis of the current integration of iron fluoride-oxide features on LSV curves, which should be directly related to the actual amount of catalytic sites in each catalyst (Fig. 4i, Supplementary Notes 6 and 7, and Supplementary Fig. 36). TOF values at $\eta = 1.6$ V initially increase upon the increase of $T_{fluorinated}$, and IFONFs-45 yields the highest TOF value of 0.2141 s$^{-1}$, which is about 5.5 times higher than that for the less defect IFONFs-15 (0.039 s$^{-1}$) (Fig. 4i and inset). The Nyquist plots (Supplementary Fig. 37) illustrate that IFONFs-45 has the smallest $R_{ct}$ of only 8.3 Ω, highlighting the ultrafast faradaic possess, which leads to a superior OER kinetics. The small $R_{ct}$ of IFONFs-45 is attributed to the synergetic effect of both rich active sites and high conductivity of nanoporous heterostructure with multiple defects. Also, the stability test was performed on IFONFs-45 and RuO$_2$ under static overpotential (Fig. 4j). The current density of IFONFs-45 exhibits negligible change after 100,000 s of testing, while RuO$_2$ remains 34.6 mA cm$^{-2}$ (86.5%) after 100,000 s durability test, indicating the excellent stability of IFONFs-45 for OER. Further investigation on the chemical composition (XPS, Supplementary Fig. 38) of IFONFs-45 after 100,000 s cycling tests further confirms the robust and stable nature toward OER. The excellent stability of IFONFs-45 should be attributed to the electrochemical stable and higher energy density of iron fluoride-oxide phase[14,20]. In particular, the IFONFs-45 shows similar high activities toward HER and OER with robust Fe$_2$O$_3$-FeF$_2$ nanodomains before and after stored in ambient atmosphere for >1 year, confirming great stability of defect-riched IFONFs-45 catalyst against ambient condition corrosion (Supplementary Fig. 39).

To investigate the water splitting performance of IFONFs-45 as bifunctional catalysts, we assembled a two-electrode system (uncompensated iR drop) using IFONFs-45 as both the anode and cathode to evaluate the overall water splitting performance in alkaline electrolyte (Supplementary Fig. 40a). With an overpotential of 1.58 V to afford 10 mA cm$^{-2}$, the IFONFs-45 reveals excellent full water splitting performance, which is superior to the benchmark of Ir/C−Pt/C couple (1.62 V). Meanwhile, the IFONFs-45 electrode could withstand continuous electrolysis over 30,000 s with less degradation than the electrolyzer containing a Ir/C−Pt/C couple at 10 mA cm$^{-2}$ (Supplementary Fig. 40b). Using an H-type cell, with an alkaline membrane for separating the anode and cathode to avoid gas mixing (Supplementary Fig. 41a), the amount of measured H$_2$/O$_2$ matches well with the calculated results, indicating the Faradic efficiencies are 100% for HER and OER with the ratio of H$_2$/O$_2$ being close to 2:1 (Supplementary Fig. 41b). Such superior electrocatalytic activity of IFONFs-45 electrode outperforms other recently reported transition metal sulfide, phosphide, nitride, carbide, selenide, and boride electrocatalysts (Supplementary Table 5).

## Discussion

We now return to extract exact bonding information of iron fluoride-oxide heterogeneous matrixes. Based on high-angle annular dark-field (HAADF) and bright-field (BF)-STEM images, the intimate contact of FeF$_2$ and Fe$_2$O$_3$ phases is verified, in which abundant interfaces can be clearly identified. As HAADF-STEM image is sensitive only to heavy atoms, only Fe atoms can be detected; all Fe atoms (white dots) in FeF$_2$ nanodomains are aligned arranged and form a tetragonal structure (Fig. 5a). The brighter spots in Fig. 5b represent either normal Fe−F or O−Fe−F columns, as BF-STEM imaging contrast is brighter for heavier atoms in atomic columns[36]. The Fe atoms (purple dots) sit at center of four F atoms (turquoise dots) (left part of enlarged dashed-box region in Fig. 5b), while the Fe atoms (purple dots) locate at center surrounded by four F−O dual atoms (turquoise and magenta dots, respectively), forming O−Fe−F bonds at interface (right part of enlarged dashed-box region in Fig. 5b). This suggests that Fe−F and O−Fe−F bonds are formed within individual atomic columns at interfaces, and FeF$_2$ chemically interacts with Fe$_2$O$_3$ by forming a Fe−F bond. Moreover, small Fe$_2$O$_3$ grains with O−Fe−F bonds are darker than the adjacent FeF$_2$ nanodomains in HAADF-STEM image[37], acting as intermediate phases, are visible overlaying iron fluoride-oxide heterogeneous nanostructure (indicated by the blue-dashed ellipses, Fig. 5a, b). The FeF$_2$ nanodomains are continuously grown on Fe$_2$O$_3$ with O−Fe−F bonds in transition regions, forming seamless interfaces. Interestingly, the location of individual F adatoms is collocated on neither Fe nor O sites within the FeF$_2$ lattice. F adatoms arising from fluorination process display mobility around the dislocation cores and edges in FeF$_2$ nanodomains; these dynamics could lead to dislocation glide and grain boundary migration[38]. Clearly, the interface involved anions disordering, is atomically abrupt and coherent with transition regions, confirming a clean and direct bonding of FeF$_2$ to Fe$_2$O$_3$ at atomic scale. Therefore, the Fe$_2$O$_3$ nanodomain itself has decent HER activity, can chemically couple with FeF$_2$ to accelerate the HER activity, while also serve as an effective support to mediate the growth of FeF$_2$, forming IFONFs heterocatalysts.

The IFONFs afford a high catalytic performance for both HER and OER. The OER activity can be attributed to Fe$_2$O$_3$, which is known as a good catalyst for OER[39]. However, the HER activity for both Fe$_2$O$_3$ and FeF$_2$ has not been reported, which raises the question what is the origin of HER activity in IFONFs. To understand it, we carried out spin-polarized density functional theory calculations with the LDA + U approach using Vienna Ab initio Simulation Package (VASP)[40] (for computational details, see Supplementary Note 8). We choose to focus on FeF$_2$, which has a rutile structure, i.e., the Fe cations are surrounded by octahedron of 6 O anions, and the O anions have a trigonal planar coordination with of 3 Fe cations. We consider the (110) surface (Fig. 5c) of FeF$_2$ crystal, which is shown by our experiments as the most common surface of FeF$_2$, in agreement with the literature[41]. To evaluate the HER activity, we get the value of H-adsorption ($\Delta G_H$) free energy on different sites through $\Delta G_H = E_{ad} + \Delta G_{vib} + TS$, where $E_{ad}$ is the adsorption energy of H, referenced to the 1/2 of energy of the H$_2$ molecule, $\Delta G_{vib}$ is the difference of vibration free energy between the adsorbed H and H$_2$ molecule, $S$ is the translation and rotation entropy of H$_2$ molecule, and $T$ is the room temperature. The $\Delta G_H$ are illustrated in Fig. 5e, and the atomic coordinates are provided in Supplementary Note 8. The site with $\Delta G_H$ closer to zero tends to have a higher catalytic activity[42]. We find that, for the perfect (stoichiometric) surface, H prefers adsorption on the Fe site, however, its $\Delta G_H$ is too high (>2 eV/H), thus the perfect surface is unlikely to be active. Defects have been shown to be active sites for other materials[43–46], hence we also consider a variety of defects, including vacancies, adatoms, substitutions (F by O), and defect complex. The most promising site appears to be the Fe−O dimer adsorbate (Fig. 5d), which binds H through the O and provides a $\Delta G_H \sim -0.15$ eV. This also suggests the presence of O is beneficial

for the HER, partially explaining why $FeF_2$–$Fe_2O_3$ hybrid has a high HER activity.

In summary, defect-enriched IFONFs are fabricated through fluorination with anodized Fe-oxide PTF. The IFONFs possessing higher electrochemical surface area can directly work as electrodes for water splitting reaction and ensure outstanding catalytic activity. This work thus will facilitate the development of newly efficient bifunctional electrocatalyst for water splitting reactions based on transitional metal foil.

## Methods

**Materials synthesis**. 3D porous Fe-oxide PTF was directly synthesized by anodic treatment of commercial Fe foil. Then, the anodized Fe-oxide PTF was reacted with fluorine vapor (from $NH_4F$) at 300–400 °C in a CVD apparatus. In CVD, $NH_4F$ sublimated at around 200 °C and decomposed to ammonia and hydrogen fluoride; upon reaction with hydrogen fluoride, Fe-oxide precursors were converted to the corresponding fluorides. The details are shown in Supplementary Information.

**Characterizations**. A JEOL 6500F SEM was used to investigate the morphology. A JEOL 2010 HRTEM was used to observe the morphologies and lattice fringes of the samples. The atomic-resolution TEM and STEM structural characterizations of IFONFs were carried out with a probe-corrected Titan G2 60–300 (FEI, USA) and Titan ChemiSTEM (FEI, USA) at acceleration voltages of 300 kV and 200 kV, respectively. The crystal structure was evaluated using XRD analysis. XPS was conducted on a PHI Quantera SXM scanning X-ray microscope. An Al anode at 25 W was used as an X-ray source with a pass-energy of 26.00 eV, 45 take-off angle, and a 100-μm beam size.

**Data availability**. The data that support the findings of this study are available from the corresponding author upon request.

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

## Acknowledgements

We acknowledge the National Natural Science Foundation of China (nos. 21603129 and U1510103), National Natural Science Foundation of Shanxi Province (no. 201601D202021), Ten Thousand Talent Program and Sanjin Scholar for finance support of this research. Y.L. acknowledges the startup support from UT Austin. We also thank Dr. Bo Chen from Rice University for assistance with XPS spectroscopy. We would also like to acknowledge Professor Boris I. Yakobson at Rice University for helpful discussions and Dr. Junjie Zhang from Scientific Instrument Center at Shanxi University for her help with ICP-MS measurement. This work used computational resources sponsored by the DOE's Office of Energy Efficiency and Renewable Energy and located at the National Renewable Energy Laboratory, and the Texas Advanced Computing Center (TACC) at UT Austin.

## Author contributions

X.F. conceived the experiment, supervised the research work, and was involved in scientific discussions. X.F. designed, carried out the syntheses, and performed electrocatalysis measurements. X.F., S.C., and J.W. performed the characterizations. Y.L. and J.S. performed first-principle calculations under the guidance of W.A.G. A.F., W.Z., S.L., and X.-M.Z. participated in the preparation of the manuscript. All the authors discussed the results and revised the paper.

## Additional information

**Competing interests:** The authors declare no competing interests.

