## [Peer Review File · Nature Communications]

Reviewers' comments:

Reviewer #1 (Remarks to the Author):

This manuscript reports the preparation of three-dimensional defect-enriched iron fluoride-oxide nanoporous thin films (IFONFs). This catalyst as a bifunctional electrocatalyst shows good performance for oxygen and hydrogen evolution reactions (OER and HER). As an excellent metal-based material, this catalyst is not competitive yet. Also, structural analysis of IFONFs is not persuadable enough. The concept of defect to improve water splitting performance has been well demonstrated by many reports. Therefore, I cannot recommend this manuscript to be published in Nature communications.

Detailed comments:

1. The Fe₂O₃ peaks for samples with long fluorinated time are not distinguishable in the XRD patterns. And the IFONFs almost have high oxygen content of 34.81-36.99 at% in XPS, it seems that this result is contradictory with the XRD result.
2. It is impossible to get the O 1s signal of adsorbed water under normal XPS conditions (ultra-high vacuum). The O 1s at high binding energy should be from surface groups of adventitious carbon.
3. There is an obvious shift of binding energy for the Fe 2p and O 1s regions of IFONFs-45 before and after 100000 s OER durability test, the author should explain the results.
4. It can be seen from the Table S4 and Table S5 that the catalytic activity of IFONFs-45 is not competitive. Especially, this catalyst is not competitive compared with other non-noble-metal based catalysts at high overpotential.
5. For HER under alkaline conditions, the authors should take the adsorption energy of H₂O molecules on catalysts for comparison into account firstly.
6. The most promising site appears to be the ad-FeO dimer adsorbate (see Fig. 5d), which binds H through the O provides low free energy of hydrogen adsorption (-0.15 eV), while v-FeF₂ has high free energy of hydrogen adsorption (> 2.0 eV). What are the active sites in IFONFs-45? 4. What is the advantage of FeF₂ for HER in alkali condition in comparison with other Fe-based compounds? How does FeF₂ affect the catalytic activity of IFONFs-45?
7. Some writing mistakes should be corrected. The references also should be revised. Please should check.

Reviewer #2 (Remarks to the Author):

The authors report excellent HER and OER activity from Iron fluoride-oxide thin films intended for rate splitting. This is a very detailed and complete study covering both characterization (detailed microscopy) and electrochemical studies. This is a high quality manuscript detailing the atomic origin of high catalytic activity backed by microscopy and modeling (DFT). Here are a few comments:

1. One important aspect is the chemical stability of the fluorination process and subsequent aging that would lead to drop in catalytic activity. To be precise, it would be interesting to report if IFONF films stored under ambient conditions still have similar efficiencies. The defect states are generally very sensitive to temperature and exposure to ambient conditions and other factors. The domain size of

Fe₂O₃-FeF₂ and the compositions could change even strong these films for longer time.

2. The fluorination methods converts Fe₂O₃ to interconnected network of FeF₂-Fe₂O₃ domains that according to this report has optimal HER and OER activity. Why no oxyfluoride formation, which should be an intermediate past before formation of FeF₂. What prevents it's formation? Are there any kinetic or thermodynamic barriers?

Reviewer #3 (Remarks to the Author):

This work reported the three-dimensional iron fluoride-oxide nanoporous films (IFONFs) through anodization/fluorination processes, and further investigated its electrochemical performance toward hydrogen evolution reaction (HER) and oxygen evolution reaction (OER). Experimental studies and first-principles calculations suggest that the surface/edge defects contribute significantly to the high-performance catalysis. I think this manuscript could be published in Nature Communications after major revision.

1. Actually, the authors prepared a highly-conductive heterostructure for electrochemical process, which have been widely reported in literatures. The authors should address the novelty of this work.

2. The authors should update the HRTEM image in order to better illustrate the lattice distances of both FeF₂ and Fe₂O₃.

3. What are the amorphous nanodomains in Fig. 2c? And why?

4. The authors declare that the presence of oxygen vacancies can decrease the adsorption energy intermediates (e.g., OH*, O*, and OOH*) at active sites and facilitate the adsorption of these intermediates. More evidences should be provided.

5. During the HER measurements, the inner resistance (IR) should be provided. The authors should state whether the polarization curves of HER is IR-corrected or not?

6. The authors should compare the HER performance of IFONFs with the typical electrocatalysts, for instances, transition metal sulfides, phosphides, nitrides, carbides, selenides and borides.

7. The number of active sites and Faradic efficiencies (FE) for both HER and OER should be calculated and provided.

8. The EIS plots should be enlarged for better comparison, since the EIS of IFONFs-45 cannot be observed.

9. The stability of both Pt and RuO₂ should be provided for better comparisons.

10. The authors should investigate the water splitting performance (electrocatalytic activity and long-term stability) of the reaction system with IFONFs-45 as both the anode and cathode. This is very important for the evaluation of electrocatalysts.

Reviewers' comments:

Reviewer #1 (Remarks to the Author):

This manuscript reports the preparation of three-dimensional defect-enriched iron fluoride-oxide nanoporous thin films (IFONFs). This catalyst as a bifunctional electrocatalyst shows good performance for oxygen and hydrogen evolution reactions (OER and HER). As an excellent metal-based material, this catalyst is not competitive yet. Also, structural analysis of IFONFs is not persuadable enough. The concept of defect to improve water splitting performance has been well demonstrated by many reports. Therefore, I cannot recommend this manuscript to be published in Nature communications.

- We thank the reviewer for constructive comments, i.e., structural aspect. These questions have been largely corrected in our revised version, and all changes are highlighted in blue throughout the revision.

For HER, IFONFs-45 outperforms most other Fe based catalysts, such as FeP nanorod arrays (*ACS Catal.*, 2014, ref. S4), FeP (*Chem. Commun.*, 2016, ref. S5), FeP₂ (*Chem. Commun.*, 2016, ref. S5), porous Ni-Fe-P@C NRs (*J. Mater. Chem. A*, 2017, ref. S6), Fe_{0.1}NiS₂ NA/Ti and NiFe-LDH NA/Ti (*Nano Res.*, 2016, ref. S7), Ni₃Fe LDHs/NF (*ACS Appl. Mater. Interfaces*, 2016, ref. S8), NiFe/NF (*Int. J. Hydrogen Energ.*, 2016, ref. S9), iron phosphide nanotubes (*Chem.–Eur. J.*, 2015, ref. S10), EG/Co_{0.85}Se/NiFe-LDH (*Energ. Environ. Sci.*, 2016, ref. S 11), and NiFe LDH-NS@DG10 (*Adv. Mater.*, 2017, ref. S 12).

The onset potential of IFONFs-45 is close to that of porous Ni-Fe-P@C NRs (*J. Mater. Chem. A*, 2017, ref. S6), while IFONFs-45 possess much smaller Tafel slope and η_{10} . Moreover, the HER performance of IFONFs-45 is also outperforms some newer, **non-Fe based catalysts**, such as 2.5H-PHNCMs (*Nat. Commun.*, 2017, ref. S14), HNDDC-100,000-1,000/Co (*Nat. Commun.*, 2017, ref. S15), 3.0 % S-CoO NRs (*Nat. Commun.*, 2017, ref. S16). Most importantly, the HER performance of IFONFs-45 also is close to some **newer, noble metal based catalysts**, such as Ru@C₂N (*Nat. Nano.*, 2017 ref. S13) and RuCo@NC (S-4) (*Nat. Commun.*, 2017, ref. S17).

For OER, IFONFs-45 outperforms most other Fe and non-Fe based catalysts, such as FeP-rGO (70 : 30)@Au (*J. Mater. Chem. A*, 2016, ref. S19), FeP NAs/CC (*ACS Catal.*, 2014, ref. S4), Fe-Ni oxides (*ACS Catal.*, 2012, ref. S21), Fe₆Ni₁₀O_x (*Angew. Chem. Int. Ed.*, 2014, ref. S25), NiFe/NF (*Int. J. Hydrogen Energ.*, 2016, ref. S9), NiFe-LDH NA/Ti (*Nano Res.*, 2016, ref. S7), EG/Co_{0.85}Se/NiFe-LDH (*Energ. Environ. Sci.*, 2016, ref. S11), NiFe LDH-NS@DG10 (*Adv. Mater.*, 2017, ref. S12), PrBa_{0.5}Sr_{0.5}Co_{1.5}Fe_{0.5}O_{5+ δ} (PBSCF-III) (*Nat. Commun.*, 2017, ref. S26), 2.5H-PH NCMs (*Nat. Commun.*, 2017, ref. S14), Ni-NHGF (*Nat. Catal.*, 2018, ref. S27), and HNDDC-100,000-1,000/Co (*Nat. Commun.*, 2017, ref. S15). Moreover, the performance of IFONFs-45 is very close to some of Fe based catalysts, such as [Ni,Fe]O (*ACS Appl. Mater. Interfaces*, 2015, ref. S18), iron phosphide nanotubes (IPNTs) (*Chem.–Eur. J.*, 2015, ref. S10), De-LNiFeP/rGO (*Energ. Environ. Sci.*, 2015, ref. S20), Ni-Fe LDH/CNT (*J. Am. Chem. Soc.*, 2013, ref. S22), FeNi-rGO LDH (*Angew. Chem. Int. Ed.*, 2014, ref. S23), NiFe LDHs (*Nat. Commun.*, 2014, ref. S24), porous Ni-Fe-P@C NRs

(*J. Mater. Chem. A*, 2017, ref. S6), Fe_{0.1}NiS₂ NA/Ti (*Nano Res.*, 2016, ref. S7), and NSPM-Ni₃FeN/NF (*ACS Appl. Mater. Interfaces*, 2016, ref. S8).

In short, the HER and OER performance of IFONFs-45 outperforms most Fe and non-Fe based catalysts. Especially, this catalyst is competitive to some newer, non-Fe based catalysts published in Nature-brand journals. These results are clear shown in *Supplementary Table 3* and *Table 4*.

As for the competitiveness of metal-based catalysts, we slightly disagree to the reviewer's point since three-dimensional defect-enriched iron fluoride-oxide nanoporous thin films (IFONFs) reported in the work shows good performance as a bifunctional electrocatalyst (OER and HER), especially in basic electrolyte that is close to practical application conditions.

There are lots of reports about defect to improve electrocatalysis performance such as HER and OER, however, defect-enriched iron fluoride-oxide nanoporous film bifunctional catalyst with high active and good stability is still rare. In this work, IFONFs with high electrical conductivity possess embedded disorder phases in crystalline lattices, and contain numerous scattered defects, including interphase boundaries, stacking faults, oxygen vacancies (V_O), and dislocations on the surfaces/interface. Defect states of IFONFs are carefully investigated. Experimental studies and first-principle calculations suggest that the surface/edge defects contribute significantly to the high performance.

We are sorry that there are lots of questions that we are not explained clear in the previous version. We have improved the manuscript (all changes highlighted in blue throughout the revision) and the replies to your comments are given below. We wish the revision fulfills your requirements to be published in *Nature Communications*.

Detailed comments:

1. The Fe₂O₃ peaks for samples with long fluorinated time are not distinguishable in the XRD patterns. And the IFONFs almost have high oxygen content of 34.81-36.99 at% in XPS, it seems that this result is contradictory with the XRD result.

➤ We appreciate the reviewer's important comments and suggestion on XRD and XPS analysis. Yes, the Fe₂O₃ peaks for IFONFs with long fluorinated time are less distinguishable in the XRD patterns. And the Fe₂O₃ peaks for IFONFs with short fluorinated time are distinguishable in the XRD patterns. As fluorinated time ($T_{\text{fluorinated}}$) prolonging, the primary diffraction peaks associated with Fe₂O₃ are declined, while both peaks for FeF₂ (110) and (101) are distinguished. During the fluorination process, the nanoporous Fe-oxide is partially converted into iron fluoride-oxide through reaction with fluorine vapor. Meanwhile, FeF₂ phase perfection increasing and FeF₂-F₂O₃ interfaces reducing occur simultaneously as more Fe₂O₃ nanodomains are transformed into FeF₂ phase.

We remeasured the XPS data, and updated data are put into the revised manuscript. We also want to point out that XPS technology is a surface detection method and can just detect several nanometers. IFONFs almost have a constant Fe content of ~19 wt% in ICP-MS, whereas F content increases from 6.76 to 16.32 at% and the surface O content decreases significantly from 45.62 to 30.38 at% in XPS, suggesting that O atoms are partial replaced by

F atoms with fluorination processing. These results agree with the XRD results.

Supplementary Table 1. Fe content determined by ICP-MS and quantitative surface analysis by XPS.

Samples	Fe loading	Surface atomic concentration (at%)			
	(wt%)	C	O	F	Fe
Fe-oxide	21.5	37.17	45.62	-	17.21
IFONFs-15	19.3	35.76	39.85	6.76	17.63
IFONFs-30	19.2	36.10	37.51	9.34	17.05
IFONFs-45	18.8	35.42	33.76	12.45	18.37
IFONFs-60	18.6	35.30	33.17	14.26	17.27
IFONFs-90	14.8	35.42	30.38	16.32	17.88

Fe-oxide PTF has a higher Fe content of 21.5 at% derived from the ICP-MS measurements. The surface O content decreases significantly from 45.62 to 30.38-39.85 at% from XPS measurements.

2. It is impossible to get the O 1s signal of adsorbed water under normal XPS conditions (ultra-high vacuum). The O 1s at high binding energy should be from surface groups of adventitious carbon.

➤ Many thanks. Yes, you are right. It is impossible to get the O 1s signal of adsorbed water under ultra-high vacuum XPS conditions. It is very possible that the O 1s at high binding energy comes from surface groups of adventitious carbon, and corresponding weak peak at ~532.1 eV is assigned to C=O groups in revised version.

We have added this discussion in page 7 line 136 to page 7 line 139 by “As revealed in **Fig. 1h**, the broad fitted peak located at around 529.9 eV corresponds to Fe-O, which is a typical peak for Fe₂O₃ that has been exposed to fluoride²⁸; while weak peak at ~532.1 eV is assigned to C=O groups²⁹.”

3. There is an obvious shift of binding energy for the Fe 2p and O 1s regions of IFONFs-45 before and after 100000 s OER durability test, the author should explain the results.

➤ We also appreciate your suggestion, we check the data, and put the right one in the revised manuscript. Due to the chemical stability of the fluorination process, the domain size of Fe₂O₃-FeF₂ and the compositions could change even strong for long time durability test. The excellent stability of IFONFs-45 should be attributed to the electrochemical stable and higher energy density of iron fluoride-oxide phase^{14, 20}.

4. It can be seen from the Table S4 and Table S5 that the catalytic activity of IFONFs-45 is not competitive. Especially, this catalyst is not competitive compared with other non-noble-metal based catalysts at high overpotential.

➤ Thank you for useful comments to improve the quality of this paper. IFONFs-45 in our manuscript favors exceptional HER properties with Tafel slope of 31 mV dec⁻¹, onset potential of mere 20 mV, and η_{10} of 47 mV. The IFONFs-45 hybrid also demonstrate excellent OER performance with an earlier onset potential of 1.39 V, lowered η_{10} of ~1.49 V, and a decreased Tafel slope (~45 mV dec⁻¹). As we stated above discussion, the HER and OER performance of IFONFs-45 outperforms most newer, **Fe and non-Fe based catalysts**. And this is much clearer shown in *Supplementary Table 3* and *Table 4*. The catalytic activity of IFONFs-45 is the best among the non-noble metal hydrogen and oxygen evolution catalysts and even approaches to the commercial noble metal based catalyst.

As for high overpotential catalytic activity, we think that the current density of catalyst at high overpotential is just one of several factors that to assess the activity of catalysts. Overall, the IFONFs-45 catalyst exhibits an excellent catalytic activity for overall water splitting as a bifunctional catalyst. The design here opens up a new simple and scalable pathway to fabricate transition metal fluorides nanoporous film as lost-cost, efficient, and robust multifunctional materials for applications.

Supplementary Table 3. Comparison of HER activity of Fe- and some non-Fe based catalysts.

Catalysts	Electrolyte	Tafel slope mV dec ⁻¹	Onset overpotential mV	η_{10} mV	Metal precursor	ref
IFONFs-45	1 M KOH	31	20	47	Fe foil	This work
FeP nanorod arrays	1 M KOH	146	86	218	Fe ₂ O ₃ nanorod arrays	ACS Catal. , 2014 ⁴
FeP	1 M KOH	75	-	194	Fe ₂ O ₃ nanowires	Chem. Commun. , 2016 ⁵
FeP ₂	1 M KOH	67	-	189	Fe ₂ O ₃ nanowires	Chem. Commun. , 2016 ⁵
Porous Ni-Fe-P@C NRs	1 M KOH	92.6	~0	79	Iron(III) nitrate nonahydrate	J. Mater. Chem. A , 2017 ⁶
Fe _{0.1} NiS ₂ NA/Ti	1 M KOH	108	-	$\eta_{20} =$ 243	Fe(NO ₃) ₃ ·6H ₂ O	Nano Res. , 2016 ⁷
NiFe-LDH NA/Ti	1 M KOH	124	-	$\eta_{20} =$ 476	Fe(NO ₃) ₃ ·6H ₂ O	Nano Res. , 2016 ⁷

Ni ₃ Fe LDHs/NF	1 M KOH	75	-	45	Fe(NO ₃) ₃ ·9H ₂ O	ACS Appl. Mater. Interfaces , 2016 ⁸
NiFe/NF	1 M KOH	112	-	139	FeSO ₄ ·7H ₂ O	Int. J. Hydrogen Energ. , 2016 ⁹
Iron phosphide nanotubes (IPNTs)	1 M KOH	59.5	31	120	(Fe(NO ₃) ₃ · 9H ₂ O	Chem.–Eur. J. , 2015 ¹⁰
EG/Co _{0.85} Se/NiFe-LDH	1 M KOH	57	240	260	Fe(NO ₃) ₃ · 9H ₂ O	Energ. Environ. Sci. , 2016 ¹¹
NiFe LDH-NS@DG10	1 M KOH	110	-	300	Fe(NO ₃) ₃ ·9H ₂ O	Adv. Mater. , 2017 ¹²
Ru@C ₂ N*	1 M KOH	38	-	17	RuCl ₃ , NaBH ₄	Nat. Nano. , 2017 ¹³
2.5H-PH NCMs*	1 M KOH	38.1	-	70	(NH ₄) ₂ MoS ₄	Nat. Commun. , 2017 ¹⁴
HNDDC-100,000-1,000/Co*	1 M KOH	93.4	-	158	Co(CH ₃ COO) ₂	Nat. Commun. , 2017 ¹⁵
3.0 % S-CoO NRs*	1 M KOH	82	-	73	CoO	Nat. Commun. , 2017 ¹⁶
RuCo@NC (S-4)*	1 M KOH	31	-	28	RuCl ₃	Nat. Commun. , 2017 ¹⁷

*Non-Fe based catalysts.

Supplementary Table 4. Comparison of OER activity of Fe- and some non-Fe based catalysts.

Catalysts	Electrolyte	Tafel slope mV dec ⁻¹	Onset overpotential V	η_{10} V	Metal precursor	ref.
IFONFs-45	1 M KOH	45	1.39	1.49	Fe foil	This work
[Ni,Fe]O	0.1 M KOH	36–48	-	300 mV	Metal chlorides	ACS Appl. Mater. Interfaces , 2015 ¹⁸
FeP-rGO (70 : 30)@ Au	1 M KOH	49.6	1.44	290 mV	Trioctylphosphine oxide (TOPO) and Trioctylphosphine (TOP)	J. Mater. Chem. A , 2016 ¹⁹
Iron phosphide nanotubes (IPNTs)	1 M KOH	43	1.48	1.52	Fe(NO ₃) ₃ ·9H ₂ O	Chem.–Eur. J. , 2015 ¹⁰
FeP NAs/CC	1 M KOH	146	86 mV	218 mV	Fe ₂ O ₃ nanorod arrays	ACS Catal. , 2014 ⁴
De-LNiFeP/rGO	1 M KOH	33.6	1.47	1.50	Fe(NO ₃) ₃	Energ. Environ. Sci. , 2015 ²⁰
Fe-Ni oxides	1 M KOH	51	-	-	Iron nitrate	ACS Catal. , 2012 ²¹
Ni-Fe LDH/CNT	1 M KOH	31	1.45	1.48	Ferrous nitrate (Fe(NO ₃) ₃)	J. Am. Chem. Soc. , 2013 ²²
FeNi-rGO LDH	1 M KOH	39	1.44	1.436	Ferrous chloride (FeCl ₃)	Angew. Chem. Int. Ed. , 2014 ²³
NiFe LDHs	1 M KOH	40	-	300 mV	Iron nitrate (Fe(NO ₃) ₃) 9H ₂ O	Nat. Commun. , 2014 ²⁴
Fe ₆ Ni ₁₀ O _x	1 M KOH	48	-	286 mV	Fe(NO ₃) ₃	Angew. Chem. Int. Ed. , 2014 ²⁵
Porous Ni–Fe–P@C NRs	1 M KOH	40	1.43	217 mV	Iron(III) nitrate nonahydrate	J. Mater. Chem. A , 2017 ⁶
Fe _{0.1} NiS ₂ NA/Ti	1 M KOH	43	-	$\eta_{100} =$ 231 mV	Fe(NO ₃) ₃ ·6H ₂ O	Nano Res. , 2016 ⁷
NSPM-Ni ₃ FeN/NF	1 M KOH	40	-	1.495	Iron nitrate Fe(NO ₃) ₃ ·9H ₂ O	ACS Appl. Mater. Interfaces , 2016 ⁸
NiFe/NF	1 M KOH	51	1.58	1.64	FeSO ₄ ·7H ₂ O	Int. J.

						Hydrogen Energ., 2016⁹
NiFe-LDH NA/Ti	1 M KOH	117	-	$\eta_{100} =$ 431 mV	Fe(NO ₃) ₃ ·6H ₂ O	Nano Res., 2016⁷
EG/Co _{0.85} Se/NiFe-LDH	1 M KOH	57	1.47	1.67	Fe(NO ₃) ₃ ·9H ₂ O	Energ. Environ. Sci., 2016¹¹
NiFe LDH-NS@DG10	1 M KOH	52	1.41	210 mV	Fe(NO ₃) ₃ ·9H ₂ O	Adv. Mater., 2017¹²
PrBa _{0.5} Sr _{0.5} Co _{1.5} Fe _{0.5} O _{5+δ} (PBSCF-III)	0.1 M KOH	52	-	358 mV	Fe(NO ₃) ₃ ·9H ₂ O	Nat. Commun., 2017²⁶
2.5H-PH NCMs*	1 M KOH	45.7	-	1.465	(NH ₄) ₂ MoS ₄	Nat. Commun., 2017¹⁴
Ni-NHGF*	1 M KOH	63	1.43	1.56	NiCl ₂ ·6H ₂ O	Nat. Catal., 2018²⁷
HNDDC- 100,000-1,000/Co*	1 M KOH	66.8	-	199 mV	Co(CH ₃ COO) ₂	Nat. Commun., 2017¹⁵

*Non-Fe based catalysts.

5. For HER under alkaline conditions, the authors should take the adsorption energy of H₂O molecules on catalysts for comparison into account firstly.

➤ We thank the reviewer for the suggestion. We have calculated the adsorption of H₂O on pure FeF₂ and the surface with FeO dimer (i.e., ad-FeO, see Fig. 5). We find that the presence of ad-FeO enhances the binding energy by ~0.05 eV, which benefits the HER under alkaline conditions.

6. The most promising site appears to be the ad-FeO dimer adsorbate (see Fig. 5d), which binds H through the O provides low free energy of hydrogen adsorption (-0.15 eV), while v-FeF₂ has high free energy of hydrogen adsorption (> 2.0 eV). What are the active sites in IFONFs-45? 4. What is the advantage of FeF₂ for HER in alkali condition in comparison with other Fe-based compounds? How does FeF₂ affect the catalytic activity of IFONFs-45?

➤ We believe that the FeO dimer adsorbed on the FeF₂ is responsible for the HER activity, while the Fe₂O₃ is the origin of OER. This applies to all the composites, including IFONFs-45. The FeF₂ itself does not have significant advantages over other Fe-based compounds for HER; however, when combined with Fe₂O₃, they demonstrate high activity for water splitting.

Our method allows for facile synthesis of these two materials together *via* an easy low temperature anodization/fluorination strategy. The iron fluoride-oxide nanoporous films (IFONFs) with open pores inherits the nanoporous nature and 3D morphology from the as-anodized Fe-oxide PTF without damaging the ordered porosity.

The heterogeneous IFONFs-45 with high electrical conductivity possess embedded disorder phases in crystalline lattices, contain numerous scattered defects, including interphase boundaries, stacking faults, oxygen vacancies (V_O), and dislocations on the surfaces/interface. Experimental studies and first-principles calculations suggest that the surface/edge defects contribute significantly to their high performance. Moreover, bare Fe_2O_3 and FeF_2 PTF present poor HER activity in terms of the largest Tafel slopes (154 and 235 $mV\ dec^{-1}$, respectively) (**Supplementary Fig. 26** and **Table 2**), highlighting the cooperative interactions between Fe_2O_3 and FeF_2 in IFONFs-45 electrocatalyst.

7. *Some writing mistakes should be corrected. The references also should be revised. Please double check.*

- We have carefully checked and corrected writing mistakes in revised manuscript, and references have been rechecked.

Reviewer #2 (Remarks to the Author):

The authors report excellent HER and OER activity from Iron fluoride-oxide thin films intended for rate splitting. This is a very detailed and complete study covering both characterization (detailed microscopy) and electrochemical studies. This is a high quality manuscript detailing the atomic origin of high catalytic activity backed by microscopy and modeling (DFT). Here are a few comments:

- We thank the reviewer for this very positive review, the main points we tried to address are perfectly summarized in the review letter. We wish the revision fulfills your requirements to be published in *Nature Communications*.

1. *One important aspect is the chemical stability of the fluorination process and subsequent aging that would lead to drop in catalytic activity. To be precise, it would be interesting to report if IFONF films stored under ambient conditions still have similar efficiencies. The defect states are generally very sensitive to temperature and exposure to ambient conditions and other factors. The domain size of Fe_2O_3 - FeF_2 and the compositions could change even strong these films for longer time.*

- We really appreciate your comment about defect-enriched IFONF films bifunctional catalyst. In this manuscript, we report defect-enriched three-dimensional (3D) iron fluoride-oxide nanoporous films (IFONFs), fabricated by anodization/fluorination process. The fluorinated time ($T_{\text{fluorinated}}$) evidently impacts extent of iron oxide to fluoride phase transformation, which in turn determines the defect states. We have studied the HER and OER performance of IFONF-45 stored under ambient conditions for more than one year, which further confirm the robustness of the hybrid catalyst. We further investigated HRTEM images of IFONFs stored under ambient conditions for more than one year. As shown in HRTEM images, the size and defect state of Fe_2O_3 - FeF_2 nanodomain didn't change after stored under ambient conditions for more than one year. The IFONFs heterogeneous matrixes still exhibit a relatively smooth surface and contain well-bonded phase junctions. The FeF_2 and Fe_2O_3 neighboring nanodomains are strongly interconnected with each other that ensure stable electrical and mechanical contact. One of the advantages of the anodization/fluorination

process is to make the FeF_2 and Fe_2O_3 neighboring nanodomains merge together at the boundary free of visible gaps, leading to nanohybrids with improved performance. This aspect has been further emphasized in the main text (page 24 and **Supplementary Fig. 39**).

We have added this discussion in page 24 line 437 to page 24 line 441 by “In particular, the IFONFs-45 shows similar high activities towards HER and OER with robust Fe_2O_3 - FeF_2 nanodomains before and after stored in ambient atmosphere for more than one year, confirming great stability of defect-rich IFONFs-45 catalyst against ambient condition corrosion (**Supplementary Fig. 39**).”

More details were provided in Supplementary Information (**Supplementary Fig. 39**).

Supplementary Figure 39 | LSV curves of IFONFs-45 electrode before (red curve) and after stored under ambient conditions for one year and four months (black dash curve) for (a) HER and (b) OER, respectively. (c, d) HRTEM images for IFONFs-45 stored under ambient conditions for one year and four months. HRTEM showing no crystalline structure change.

2. The fluorination methods converts Fe_2O_3 to interconnected network of FeF_2 - Fe_2O_3 domains that according to this report has optimal HER and OER activity. Why no oxyfluoride formation, which should be an intermediate past before formation of FeF_2 . What prevents its formation? Are there any kinetic or thermodynamic barriers?

- This concern is reasonable. A recent report by Nanda et al. showed at lower fluorination temperatures (<275 °C) during fluorination reaction, a thin F-rich layer formed on the surface of the Fe₂O₃ particles (*ACS Nano* **9**, 2530-2539 (2015)). This amorphous or semicrystalline shell had a nominal stoichiometry of oxyfluoride phase. In our cases, we use ammonium fluoride (NH₄F) as fluorine gas to convert Fe-oxide nanoporous film to defect enriched iron fluoride-oxide nanoporous films (IFONFs) under relative high fluorination temperature (300-400 °C). Under this condition, F penetrated into the Fe₂O₃ phase, at the same time, the FeF₂ phase began to crystallize. Finally, interconnected F-rich (FeF₂) and O-rich (Fe₂O₃) domains coexisted within the nanoporous film, rather than oxyfluoride nanodomain formation. Therefore, the direct conversion of the oxide to the fluoride phase without intermediate of oxyfluoride during fluorination at relative high fluorination temperatures (300-400 °C), which is consistent with prior literature reports.

Moreover, according from first principle-based thermodynamic calculations by Vincent and co-workers, iron oxide-fluoride has higher formation energy conversion to oxyfluoride than that of iron fluoride. FeOF is traditionally synthesized through a solid-state reaction of FeF₃ and Fe₂O₃ in an argon atmosphere at 950 °C. There may be kinetic barriers hindering the formation of FeOF from Fe₂O₃+F₂, as the reviewer suggested (*Phys. Rev. B* 2013, 87, 094118.). FeOF has higher entropic stabilization energy (*ACS Nano* **9**, 2530-2539 (2015)).

In our cases, the iron oxide-fluoride formed at under relative high fluorination temperature prevents the formation of oxyfluoride. FeF₂ phase was the more stable than oxyfluoride, and did not form FeOF even annealing FeF₂ in air at 300 °C (*J Electrochem Soc* 156, A407-A416 (2009).). So the FeF₂ phase form with fluorination at 300-400 °C could not converted to FeOF through reaction with fluorine vapor. Therefore, defect enriched IFONFs are fabricated through fluorination with anodized Fe-oxide PTF without formation of oxyfluoride phase.

Reviewer #3 (Remarks to the Author):

This work reported the three-dimensional iron fluoride-oxide nanoporous films (IFONFs) through anodization/fluorination processes, and further investigated its electrochemical performance toward hydrogen evolution reaction (HER) and oxygen evolution reaction (OER). Experimental studies and first-principles calculations suggest that the surface/edge defects contribute significantly to the high-performance catalysis. I think this manuscript could be published in *Nature Communications* after major revision.

- We appreciate reviewer's evaluation of our manuscript. Our responses to the comments are described as below. We wish the revision fulfills your requirements to be published in *Nature Communications*.

1. Actually, the authors prepared a highly-conductive heterostructure for electrochemical process, which have been widely reported in literatures. The authors should address the novelty of this work.

- We really appreciate your comment. The novelty of this work lies at least two aspects:

defect-enriched iron fluoride-oxide electrocatalyst and easy low temperature anodization/fluorination preparation strategy. In this work, 3D iron fluoride-oxide nanoporous films (IFONFs) were fabricated by anodization/fluorination process. The catalyst exhibits an excellent catalytic activity for overall water splitting as a bifunctional catalyst. The excellent catalytic performance is attributed to the structure and composition of the catalyst. The 3D nanostructure with numerous scattered defects provides more active sites, owing to the high specific surface area, and promotes gas production and bubble release. IFONFs-45 shows a good electrical conductivity that decreases the resistance of the catalytic system. This low-cost, stable IFONFs-45 is directly used as a bifunctional catalyst in a two-electrode alkaline electrolyzer for hydrogen and oxygen production. This work thus will **facilitate the development of newly efficient bifunctional electrocatalyst for water splitting** reactions based on transitional metal foil.

2. *The authors should update the HRTEM image in order to better illustrate the lattice distances of both FeF_2 and Fe_2O_3 .*

➤ We really appreciate your comment. We have updated the HRTEM image of **Fig. 2c** with better illustrate the lattice distances of both FeF_2 and Fe_2O_3 . Moreover, as **Fig. 3** in previous vision was not clear enough, we also edited the figure to a higher resolution.

3. *What are the amorphous nanodomains in Fig. 2c? And why?*

We apologize if the sense of this sentence is not clear. According to reference 31(Adv. Mater. 2015, 27, (20), 3208-3215.), the amorphous rims in **Fig. 2c** are the small amount initiation layer remnants or other etching artifacts. The amorphous nanodomains (marked by white dashed line) are the residuum of as-formed Fe-oxide nanoporous films fabricated through anodic treated of Fe foil. Most Fe-oxide nanodomains are therefore crystallized into a mixed FeF_2 and Fe_2O_3 phase by fluorinated in Ar at 300-400 °C. While a small amount of Fe-oxide amorphous nanodomain still exist in FeF_2 - Fe_2O_3 hybrid. We modified the sentence and the meaning should now be more clear (see page 11 main text).

We have added this discussion in page 11 line172 to page 11 line 182 by “**Fig. 2c** represents the zoom-in imaging of a rectangular region in **Fig. 2b** (marked with yellow dotted line), which indicates that the nanopores possess a smooth space morphology with amorphous rims containing a small amount initiation remnant or other etching artifacts³¹. The amorphous nanodomain (marked by white dashed line, **Fig. 2c and Supplementary Fig. 13b**) is the residuum of as-formed Fe-oxide nanoporous films. Notably, besides amorphous nanodomains, the lattice fringes with spacing of ~0.325 and 0.248 nm ascribed to FeF_2 (110) and Fe_2O_3 (400), respectively, are clearly visible, indicating crystallinity of iron fluoride-oxide nanostructure. The nanopores, together with fine nanograins of iron fluoride-oxide and thin amorphously shells are uniformly distributed in the porous network, ensuring IFONFs-45 has more accessible sites for electrochemical reactions.”

4. *The authors declare that the presence of oxygen vacancies can decrease the adsorption energy intermediates (e.g., OH^* , O^* , and OOH^*) at active sites and facilitate the adsorption of these intermediates.*

More evidences should be provided.

- We thank the reviewer for the comments. We have performed additional calculations for OH adsorption on perfect and defective Fe_2O_3 (with oxygen vacancy), as shown in the Figure below. We selected (100) surface because it is the most stable and active surface (ref: Zhang et al. Ref. DOI: 10.1021/acs.jpcc.6b10553). We used a 2×2 supercell with $\sim 15 \text{ \AA}$ vacuum layer, and the spin polarized DFT+U formalism with the U value of 4.3 eV for Fe (Ref: DOI: 10.1021/ja301567f). The energy cut-off for the plane waves was set to 400 eV, and all atomic positions were fully relaxed with a Γ point until the final force on each atom was less than 0.01 eV/\AA . Indeed, our calculations show that the adsorption energy is facilitated by 0.99 eV at the O vacancy, supporting our claim in the paper.

We have added this discussion in page 15 line 245 to page 15 line 250 by “The presence of V_{O} can decrease the adsorption energy of intermediates (*e.g.*, OH^* , O^* , and OOH^*) at active sites and facilitate the adsorption of these intermediates³⁵. To validate the above assumption, we have performed additional calculations for OH adsorption on perfect and defective Fe_2O_3 (with oxygen vacancy), as shown in the **Supplementary Fig. 19**. Theoretical calculations demonstrated that the adsorption energy is facilitated by 0.99 eV at the O vacancy, which is also advantageous for electrocatalysis.”

Supplementary Figure 19 | Top and side views of OH adsorption on (a) the perfect Fe_2O_3 (100) surface, (b) the surface with one O vacancy. The OH groups are marked by blue circles.

5. During the HER measurements, the inner resistance (IR) should be provided. The authors should state whether the polarization curves of HER is IR-corrected or not?

- We added related descriptions to the supplementary materials following the reviewer's suggestion. The polarization curves of HER are without IR-corrected.

6. The authors should compare the HER performance of IFONFs with the typical electrocatalysts, for instances, transition metal sulfides, phosphides, nitrides, carbides, selenides and borides.

- We really appreciate your comment. We have added the HER and OER performance of transition metal sulfides, phosphides, nitrides, carbides, selenides and borides compare with IFONFs in **Table S5**.

We have added this discussion in page 25 line 459 to page 25 line 461 by “Such superior electrocatalytic activity of IFONFs-45 electrode outperforms other recently reported transition metal sulfides, phosphides, nitrides, carbides, selenides and borides electrocatalysts (Supplementary Table 5).”

Supplementary Table 5. Comparison of HER and OER activity data among various catalysts.

Catalysts		Reactions	Tafel slope mV dec ⁻¹	η_{10} mV	η_{100} mV	E_{onset} mV	ref
Fluoride	IFONFs-45	HER	31	47	199	20	This work
		OER	45	1.49	1.69	1.39	
Borides	FeB ₂	HER	87.5	61	200*	20	28
		OER	52.4	296	1.63*	1.48	
	Fe ₂ B	HER	102.4	138	250*	-	28
		OER	78.7	-	1.65*	1.54	
Sulfides	Fe _{0.1} NiS ₂ NA/Ti	HER	108	250*	350*	-	7
		OER	43	205*	231	-	
	(Ni _{0.75} Fe _{0.25})Se ₂	HER	NA	NA	NA	NA	29
		OER	47.2	η_{35} =255	277	-	
Phosphides	FeP NAs/CC	HER	146	218	275*	86	4
		OER	NA	NA	NA	NA	
	FeP NWs	HER	75	194	NA	-	5
		OER	NA	NA	NA	NA	
	FeP ₂ NWs	HER	67	189	NA	-	5
		OER	NA	NA	NA	NA	
	Iron phosphide nanotube (IPNTs)	HER	59.5	120	180*	31	10
		OER	43	1.52	1.57*	1.48	
Nitrides/ Selenides	Nanoporous Fe _x N	HER	NA	NA	NA	NA	30
		OER	44.5	-238	1.53*	-	
	Co ₃ FeN _x	HER	94	23	147	-	31
		OER	46	η_{20} =222	253	-	

	Ni ₃ FeN-NPs	HER	42	158	310*	-	32
		OER	46	280	1.62*	-	
Carbides	Fe ₃ C@NG800-0.2	HER	NA	NA	NA	NA	33
		OER	62	1.59	1.61*	-	
	IP-IC@SWNT(P)	HER	87.6	301	NA	-	34
		OER	NA	NA	NA	NA	
	Fe@C-NG/NC NTs	HER	NA	NA	NA	NA	35
		OER	163	1.68	η_{20} =1.72*	-	

* The value is calculated from the curves shown in the literatures.

7. The number of active sites and Faradic efficiencies (FE) for both HER and OER should be calculated and provided.

- We really appreciate your comment. We have conducted the number of active sites and Faradic efficiencies (FE) for both HER and OER measurements. The results have been included in the supporting information.

For The number of active sites of HER:

We have added this discussion in page 21 line 356 to page 21 line 359 by “The number of active sites was quantified by an electrochemical method (**Supplementary Fig. 30**). The results show that the number of active sites for IFONFs-45 is 1.09×10^{-6} mol, much larger than that of IFONFs-30 (1.94×10^{-7} mol) and IFONFs-60 (2.26×10^{-7} mol).”

More details were provided in Supplementary Information (**Supplementary Fig. 30**) by “Since the difficulty in attributing the observed peaks to a given redox couple, the number of active sites should be proportional to the integrated charge over the CV curve. Assuming a one-electron process for both reduction and oxidation, the upper limit of active sites (n) for IFONFs-45 could be calculated according to the follow equation:

$$n = Q/2F$$

where F and Q are the Faraday constant and the whole charge of CV curve, respectively.

$$n_{\text{Pt/C}} = \frac{0.2776}{2 * 96485} = 1.44 * 10^{-6}$$

$$n_{\text{IFONFs-30}} = \frac{0.0374}{2 * 96485} = 1.94 * 10^{-7}$$

$$n_{\text{IFONFs-45}} = \frac{0.2096}{2 * 96485} = 1.09 * 10^{-6}$$

$$n_{\text{IFONFs-60}} = \frac{0.0436}{2 * 96485} = 2.26 * 10^{-7}$$

Supplementary Figure 30 | CVs of IFONFs-45 and Pt/C in 1.0 M KOH (pH 14) with a scan rate of 50 mV s⁻¹ in the region of -0.2 to 0.6 V vs RHE.

For the number of active sites of OER:

We have added this discussion in page 23 line 416 to page 23 line 419 by “To further unravel the intrinsic activities, the number of active sites and TOF for OER were calculated on basis of the current integration of iron fluoride-oxide features on LSV curves, which should be directly related to the actual amount of catalytic sites in each catalyst (**Supplementary Note 6-7, Supplementary Fig. 36 and Fig. 4i**).”

More details were provided in Supplementary Information (**Supplementary Figure 36**) by “A linear plot between the oxidation currents for redox species and scan rates can be derived from cyclic voltammograms, and the corresponding slopes can be obtained from the linear plots. The quantity of active species (m) is calculated based on the formula: slope = $n^2F^2m/4RT$, where n is the number of electrons transferred, which is denoted as 1 in order to achieve the upper limit in the concentration of active sites, F is the Faradic constant (96485 C mol⁻¹), m is the number of active species, and R and T are the ideal gas constant (8.314 J mol⁻¹ K⁻¹) and absolute temperature (298 K), respectively. The results showed that the number of active sites for IFONFs-45 is 2.59×10^{-8} mol, much larger than that of IFONFs-30 (1.26×10^{-8} mol) and IFONFs-60 (1.50×10^{-8} mol).

$$m_{Fe-oxide} = \frac{0.00406 * 4 * 8.314 * 298}{96485 * 96485} = 4.31 * 10^{-9} \text{ mol}$$

$$m_{IFONFs-15} = \frac{0.00646 * 4 * 8.314 * 298}{96485 * 96485} = 6.88 * 10^{-9} \text{ mol}$$

$$m_{IFONFs-30} = \frac{0.01181 * 4 * 8.314 * 298}{96485 * 96485} = 1.26 * 10^{-8} \text{ mol}$$

$$m_{IFONFs-45} = \frac{0.02431 * 4 * 8.314 * 298}{96485 * 96485} = 2.59 * 10^{-8} \text{ mol}$$

$$m_{IFONFs-60} = \frac{0.0141 * 4 * 8.314 * 298}{96485 * 96485} = 1.50 * 10^{-8} \text{ mol}$$

$$m_{IFONFs-90} = \frac{0.00791 * 4 * 8.314 * 298}{96485 * 96485} = 8.42 * 10^{-9} \text{ mol}$$

Supplementary Figure 36 | (a, c, e, g, I and k) Cyclic voltammograms of Fe-oxide PTF and IFONFs hybrids under different scan rates increasing from 10 to 100 mV s⁻¹ in 1.0 M KOH. (b, d, f, h, j and l) Linear relationship of the peak current for the oxidation wave at the scan rate.

For Faradic efficiencies (FE) for both HER and OER:

We have added this discussion in page 25 line 450 to page 25 line 453 by “Using an H-type cell, with an alkaline membrane for separating the anode and cathode to avoid gas mixing (Supplementary Fig. 41a), H₂ and O₂ with a predicted ratio of 2 :1 are detected, and the amount of measured H₂ and O₂ matches well with the calculated results, indicating a nearly 100% Faradaic efficiency (Supplementary Fig. 41b).”

More details were provided in Supplementary Information (Supplementary Figure 41) by “The Faradic efficiencies (FEs) for both processes were calculated by comparing the amount of experimentally quantified gas with theoretically calculated gas. The rough agreement of both values suggests the FEs are 100% for HER and OER with the ratio of H₂ and O₂ being close to 2:1.”

Supplementary Figure 41 | (a) Image showing the evolution of H₂ and O₂ gas on IFONFs-45 in a well-sealed H-type cell. The IFONFs-45 samples are sealed with parafilm using electric wires connected with electrochemical workstation. (b) Experimental and theoretical amounts of H₂ and O₂ by the IFONFs-45 electrode at a fixed current density of 40 mA cm⁻².

8. The EIS plots should be enlarged for better comparison, since the EIS of IFONFs-45 cannot be observed.

➤ We thank the reviewer for the suggestion. Enlarged EIS plots are added in the revised manuscript.

9. The stability of both Pt and RuO₂ should be provided for better comparisons.

- We completely agree with your opinion. We have provided the long-term stability testing experiment of Pt and RuO₂ in HER and OER conditions, respectively, which were added in the main text and **Supplementary Fig. 31**.

We have added this discussion in page 21 line 375 to page 22 line 378 by “In contrast, the current density of Pt/C decreases from 82 to 74.1 mA cm⁻² for 30000 s of continuous operation (**Supplementary Fig. 31**). This confirms the better stability of IFONFs-45 than that of Pt/C.”

Supplementary Figure 31 | Time-dependent current density curve of Pt/C at a fixed overpotential of -70 mV to drive 82 mA cm⁻² for 30000 s in 1.0 M KOH.

10. The authors should investigate the water splitting performance (electrocatalytic activity and long-term stability) of the reaction system with IFONFs-45 as both the anode and cathode. This is very important for the evaluation of electrocatalysts.

- We really appreciate your comment. We have added electrocatalytic activity and long-term stability of IFONFs-45 as both the anode and cathode in revised edition.

Remarkably, the IFONFs-45 hybrid exhibited outstanding performance for overall water splitting with an overpotential of 1.58 V to afford 10 mA cm⁻², which surpasses that of the Ir/C–Pt/C couple (1.62 V) for sufficiently high overpotentials (large current densities). Meanwhile, the IFONFs-45 electrode could withstand continuous electrolysis for at least 30000 s with less degradation than the benchmarking combination of the Ir/C–Pt/C couple at a current density of 10 mA cm⁻² (**Supplementary Fig. 40**).

We have added this discussion in page 25 line 442 to page 25 line 450 by “To investigate the water splitting performance of IFONFs-45 as bifunctional catalysts, a two-electrode setup (uncompensated iR-drop) using 3D IFONFs-45 as both the anode and cathode was assembled to investigate its performance for overall water splitting in 1.0 M KOH solution (**Supplementary Fig. 40a**). The IFONFs-45 hybrid exhibited outstanding

performance for overall water splitting with an overpotential of 1.58 V to afford 10 mA cm^{-2} , which surpasses that of the Ir/C–Pt/C couple (1.62 V) for sufficiently high overpotentials (large current densities). Meanwhile, the IFONFs-45 electrode could withstand continuous electrolysis for at least 30000 s with less degradation than the benchmarking combination of the Ir/C–Pt/C couple at a current density of 10 mA cm^{-2} (Supplementary Fig. 40b).”

More details were provided in Supplementary Information (Supplementary Figure 40).

Supplementary Figure 40 | (a) Polarization curves of IFONFs-45 (+ and -), Pt/C (+)//Pt/C (-), and Ir/C (+)//Pt/C (-) for overall water splitting in a two-electrode configuration (not iR -corrected). (b) Chronopotentiometry curves of IFONFs-45 and Ir/C (+)//Pt/C (-) under a current density of 10 mA cm^{-2} without iR correction. All experiments were carried out in 1.0 M KOH.